# Governing principles of hydration of mixed proton conducting Co-based double perovskites

Ragnar Strandbakke [1,2] ✉, Sebastian Lech Wachowski [3], Maria Balaguer [4], Lasse Vines[5], Thomas Neset Sky[5], Iga Szpunar [3,6], Patricia A. Carvalho[1], Aleksandra Mielewczyk-Gryń [3], Magnus H. Sørby[7], Maria Gazda [3], Jose M. Serra [4] & Truls Norby [2] ✉

Proton ceramic electrochemical cells PCECs hold promise for efficient, sustainable production and use of hydrogen. The positive electrodes are mixed proton conducting perovskites that facilitate water splitting and oxygen reduction, but the factors that determine the protonation are poorly understood. Here, we establish the governing principles of protonation through a study of hydration of 45 double perovskites with the general formula $A^IA^{II}Co_2O_{6-\delta}$, having Ba or Ba+Sr on $A^I$ and a mix of rare earths (Y and lanthanides $Ln$ = La, Pr, Nd, Sm, Gd, Dy, Tb, Lu) on $A^{II}$. We show how hydration is coupled to the A-site basicity and disorder as well as population of electron holes in the Co-O bond (Co oxidation state), promoted by a closed or semi-closed $Ln$ 4 f shell, i.e., $Ln$ = La, Gd, Lu.

There is an increasing interest in the hydration of mixed proton-electron conductors (MPECs) as positive electrodes (positrodes) of proton ceramic electrochemical cells (PCECs)[1–10], but quantification of the minor proton concentrations, as well as description and experimental verification of how the local chemical and electronic structure affects hydration, are ambiguous. Atomistic modelling shows that proton and hydroxide affinities are related to the ionicity of the cation-oxygen bonds and that proton affinity dominates over hydroxide affinity in hydration thermodynamics[11]. Ionic bonds with complete cation-to-oxygen electron transfer are thus pivotal for hydration and significant proton concentrations.

In this study, we experimentally show how the A-site cation influences the charge transfer in double perovskites with $A^I$ and $A^{II}$ ordering and how the A-site composition is directly linked to proton stability in the structures. We show how lanthanides ($Ln$) with closed (La or Lu) or semi-closed (Gd) 4f shells situated on the $A^{II}$-site influence electron states in the Co-O bond and – through an extensive experimental matrix – how these particular compositions promote hydration. Furthermore, our results reveal how hydration may be over- or underestimated through additional oxidation or reduction as the crystal structure equilibrates during the hydration process.

Hydration of Ba-based mixed conducting transition metal (TM) perovskites has been investigated in a series of studies, focusing on how the electron structure of the TM-O bond influences hydration[12–17]. There has been a focus on cation acidity and basicity, repulsion between protons and electron holes, and how materials with electron holes as dominating positive defects are reduced through what becomes hydrogenation[12]. Moreover, if an electron hole is partly or fully transferred from the TM to oxygen – so-called negative charge transfer – proton stability decreases. Trends of negative charge transfer versus TM-O bond lengths and TM-O-TM angles have been established using Density Functional Theory (DFT) calculations and

[1]SINTEF AS, Oslo, Norway. [2]Department of Chemistry, Centre for Materials Science and Nanotechnology, University of Oslo, Oslo, Norway. [3]Institute of Nanotechnology and Materials Engineering, Faculty of Applied Physics and Mathematics, and Advanced Materials Centre, Gdańsk University of Technology, Gdańsk, Poland. [4]Instituto de Tecnología Química (ITQ), Consejo Superior de Investigaciones Científicas-Universitat Politècnica de València, Valencia, Spain. [5]Department of Physics, Centre for Materials Science and Nanotechnology, University of Oslo, Oslo, Norway. [6]Wallenberg Initiative Materials Science for Sustainability, Department of Chemistry and Chemical Engineering, Chalmers University of Technology, Gothenburg, Sweden. [7]Department for Neutron Materials Characterization, Institute for Energy Technology, Kjeller, Norway. ✉e-mail: ragnar.strandbakke@sintef.no; truls.norby@kjemi.uio.no

X-ray Absorption Spectroscopy[13–15], concluding that longer TM-O bonds and increasing deviation from TM-O-TM = 180° increase the ionic character of perovskites and thus oxide ion basicity and, in turn, hydration.

Many Fe- and Co-based perovskites exhibit considerable hydration[2,4,6,18]. In this work, hydration has been systematically studied in 45 different $A^I A^{II} Co_2 O_{6-\delta}$ double perovskites with mainly Ba on $A^I$, a range of rare earths and lanthanides on $A^{II}$, and additional substitutions of Sr or lanthanides on the $A^I$ site and Fe or Ti for Co on the B site. Figure 1 shows an orthorhombic a2b2c *Pmmm* unit cell of $BaGd_{0.8}La_{0.2}Co_2O_{6-\delta}$ (BGLC82) with Ba in $A^I$, Gd and La in $A^{II}$, Co in B, and with seven oxygen sites refined from synchrotron radiation powder X-ray diffraction (SR-PXD) and Neutron Powder Diffraction (NPD)[19]. Common for all compositions is that oxygen vacancies are located and tend to order within the *Ln* layer, resulting in a doubled b-axis and orthorhombic structure[19,20]. Oxygen vacancies at O7 render the Co positioned above it in pyramidal coordination with oxidation states +2 and +3 (labelled $Co_{red}$). Co above the O3 position is in octahedral coordination with oxidation states +3 and +4 (labelled $Co_{ox}$)[21].

We report experimental evidence for the effect of A-site cation basicity – reflected by their individual Allred–Rochow electronegativity through increasing *Ln* ionic radius –, TM-O-TM bond-character, and B-site oxidation state on proton concentration. Most importantly, we show that these factors alone are not sufficient: hydration in Co-based double perovskites requires that the *Ln* 4f-shell must be closed (empty La or full Lu) or semi-closed (half-full Gd).

The results show that the trends reported for Fe-based materials also apply to Co-based compositions, but also that proton concentrations are easily over- or underestimated from thermogravimetric experiments: The hydration process is coupled to slow A-site cation disorder that alters oxidation equilibria, resulting in additional weight gain from oxygen uptake in humid, oxidising atmosphere.

## Results

### Hydration of $A^I A^{II} Co_2 O_{6-\delta}$ as a function of compositional A-site matrix

Figure 2a shows a selection of compositions screened for hydration using TG at 300 °C in dry and humidified air (data partly adapted from a previous work[20]). Only compositions with closed- or semi-closed *Ln* 4f-shell on the $A^{II}$-site (La, Gd, Lu) show significant hydration. Figure 2b displays hydration for all compositions (with abbreviations listed in Supplementary Table 1), including multiple substitutions on $A^I$, $A^{II}$, and B versus the averaged Ionic Radius[22] Ratio

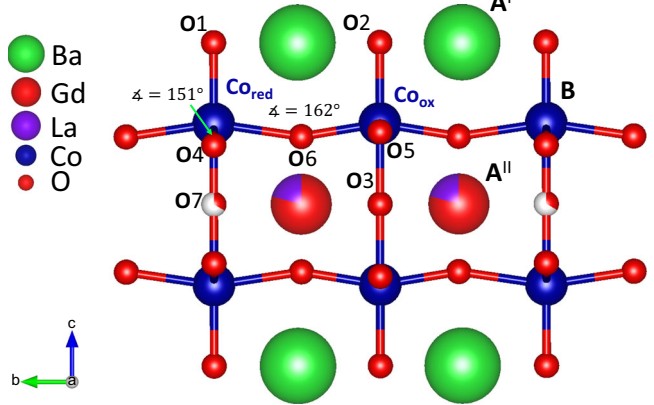

**Fig. 1 | The structure of BGLC82 *Pmmm*.** Seven O positions are identified: O1 and O2 in the Ba-O layer, O4, O5, and O6 in the Co-O layer, and O7 and O3 in the Ln-O layer. O vacancies are refined to O7[19]. Two Co positions are discerned; $Co_{red}$ in plane pyramidal coordination and $Co_{ox}$ in octahedral coordination[21].

(IRR) between the cations on $A^I$ and $A^{II}$ sites. It is evident that proton concentrations diminish with the introduction of *Ln* with open 4f-shells. Mixtures of more than two closed-shell *Ln* on $A^{II}$ and substitution of Sr for Ba on $A^I$ also suppress hydration. Some of the compositions – with Sr on $A^I$ or high concentrations of La on $A^{II}$ – crystallise in a cubic $ABO_3$ structure, also suppressing hydration. Acceptor doping of Sr for *Ln*, giving an overall A-site valence below +2.5, increases the concentration of oxygen vacancies and, hence, the degree of hydration, as seen for cubic $BaLa_{0.75}Sr_{0.25}Co_2O_{6-\delta}$, which shows a six-fold higher hydration than cubic BLC. Substituting 10% Ti or Fe for Co lowers the mass gain during hydration. All in all, even though proton concentrations for the hydrating compositions are relatively low – up to around 1 mol% under oxidising conditions – the results show a clear trend: Significant hydration is seen only for compositions where the lanthanides have either empty (La), half-full (Gd), or full (Lu) 4f-shells.

### X-ray absorption spectroscopy (XAS): effect of Co electron structure and oxidation state

The indirect effect of the *Ln* 4 f shell on the Co-O electron structure is illustrated in Fig. 2c–h, where we present X-ray Absorption Spectra with spectral lines from 526 to 532 eV, representing the pre-edge structure of the oxygen K-edge in hydrating $Ba_{0.5}La_{0.5}CoO_3$ (BLC) and $BaGdCo_2O_{6-\delta}$ (BGC), non-hydrating BPC and BNC (c), and $BaGd_xLa_{1-x}Co_2O_{6-\delta}$ (x = 0, 0.3, 0.5, 0.8, and 1) (d–h) after treatments in dry and humid atmospheres. The first maximum at ~528 eV represents the overlap of Co3d $t_{2g}$ and O2p orbitals, while the second maximum at ~530 eV represents Co3d $e_g$ and O2p overlap[23]. The first peak position shifts along the *Ln* series (Fig. 2c), reflecting how the Co3d $t_{2g}$ and O2p energy scales with *Ln* radius[22]. The average Co oxidation state depends on the *Ln* radius[20,24–26], which, in turn, affects the bond energy levels.

The spectral weight at ~528 eV indicates electronic depopulation of the Co3d $t_{2g}$-O2p bond[27]. Non-hydrating BNC and BPC show higher intensity, reflecting more electron holes, while the closed-shell BLC and BGC exhibit lower intensity and thus more electrons and/or less covalent character for Co3d $t_{2g}$-O2p.

Figure 2d–h show how the electron structure in the closed-shell compositions shifts before and after exposure to wet atmosphere in air at 300 °C, revealing a significant increase in the intensity of the Co3d $t_{2g}$-O2p signal for BLC, $BaGd_{0.3}La_{0.7}Co_2O_{6-\delta}$ (BGLC37), BGLC82 and BGC, after exposure to $H_2O$. The absence of signal for BGLC37 and BGLC82 in the dry state represents either a fully ionic bond, with no Co3d $t_{2g}$-O2p overlap, or an overlapping bond, with only occupied states in a low-spin configuration. The $e_g$ orbital is directed in the ab plane between Co and oxygen, while $t_{2g}$ is directed diagonally towards the A-site cation[28]. Repulsion from protons situated at the nearby oxide ion causes hole transfer to $t_{2g}$ and a decrease in the Co3d $e_g$-O2p hole population upon hydration. This decrease is higher for BGLC37 and BGLC82 than for BGC and BLC, indicating lower proton concentrations in the latter two.

The exception from the closed-shell hydration trend is BGLC55, which shows no hydration from TG (Fig. 2b), high spectral weight for Co3d $t_{2g}$-O2p, and no increase in $t_{2g}$ intensity after exposure to $H_2O$ (Fig. 2f). The symmetrical BGLC55 may be seen as a special case with respect to electronic structure, displaying elongated a- and b-unit cell parameters[19] and more unoccupied states in the Co2d $t_{2g}$-O2p orbitals, reflecting a high-spin configuration.

### Effects of structure: diffraction studies

Most compositions adopt A-site order, where Ba and *Ln* prefer $A^I$ and $A^{II}$ positions, respectively, and the structures can be refined to either tetragonal *P4/mmm* or orthorhombic *Pmmm* structures (Fig. 1). One exception is $Ba_{0.5}La_{0.5}CoO_3$, which can be cubic BLC(C) with an average Co oxidation state +3.43, or A-site-ordered orthorhombic

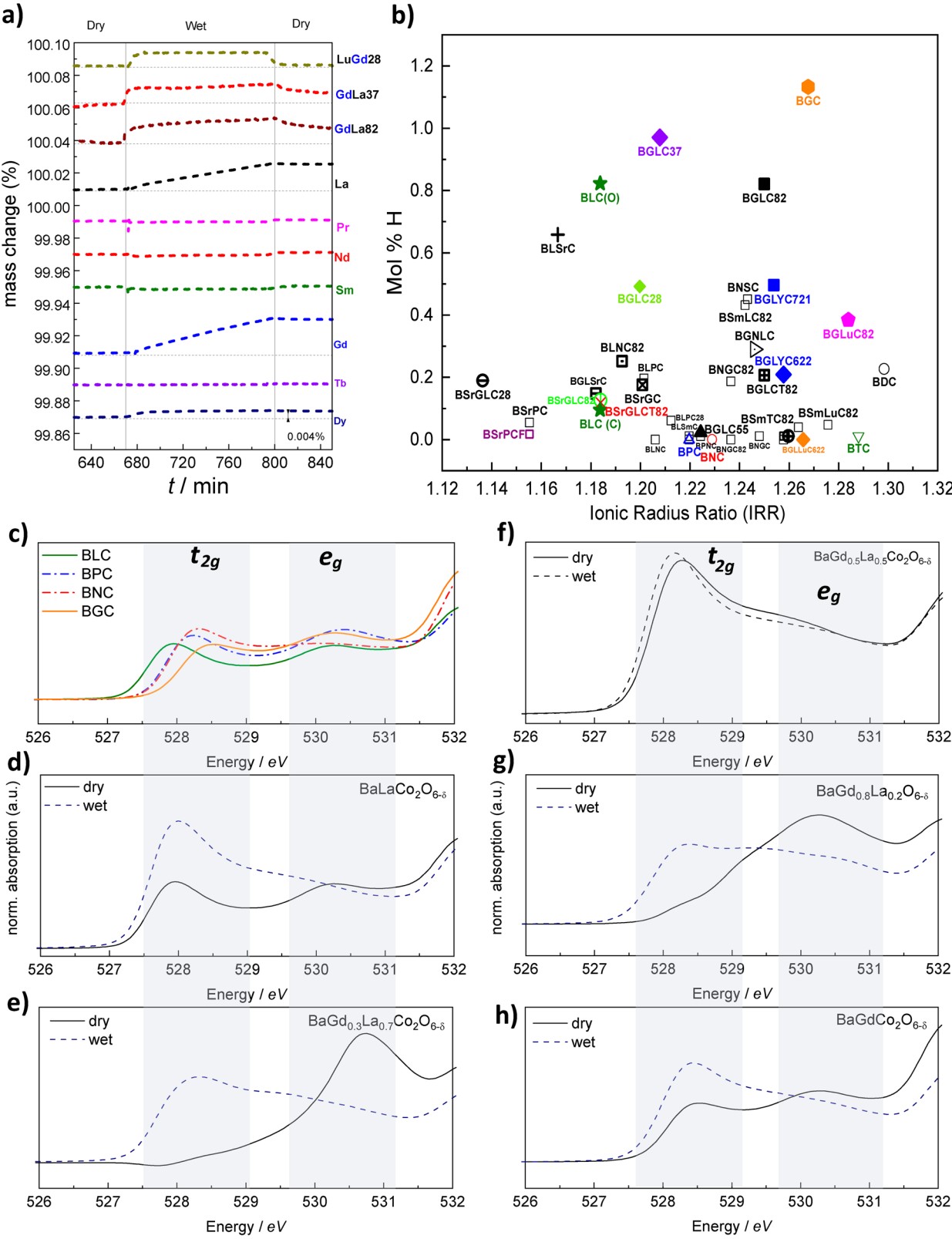

**Fig. 2 | Hydration of Ba$Ln$Co$_2$O$_{6-\delta}$. a** Hydration in a selected range of Ba$Ln$Co$_2$O$_{6-\delta}$ compositions at 300 °C in air. **b** Proton concentrations in all compositions estimated by initial weight gain attributed to hydration in a humid (H$_2$O) atmosphere at 300 °C versus A$^I$/A$^{II}$ Ionic Radius Ratio. See also Supplementary Table 1 for literature references and details on sample compositions and weight gains. **c** X-ray absorption spectra of oxygen K-edge in the Co3d-O2p bands region for open-shell BPC and BNC and closed-shell BLC and BGC (data adapted from ref. 34). **d**–**h** XAS spectra of Co3d-O2p band region for closed-shell BLC, BGLC37, BGLC55, BGLC82 and BGC in dry (solid lines) and hydrated (dashed lines) state. Shaded regions in (**c**–**h**) are typical energy ranges for $t_{2g}$ and $e_g$ states. Source data are provided as a Source Data file.

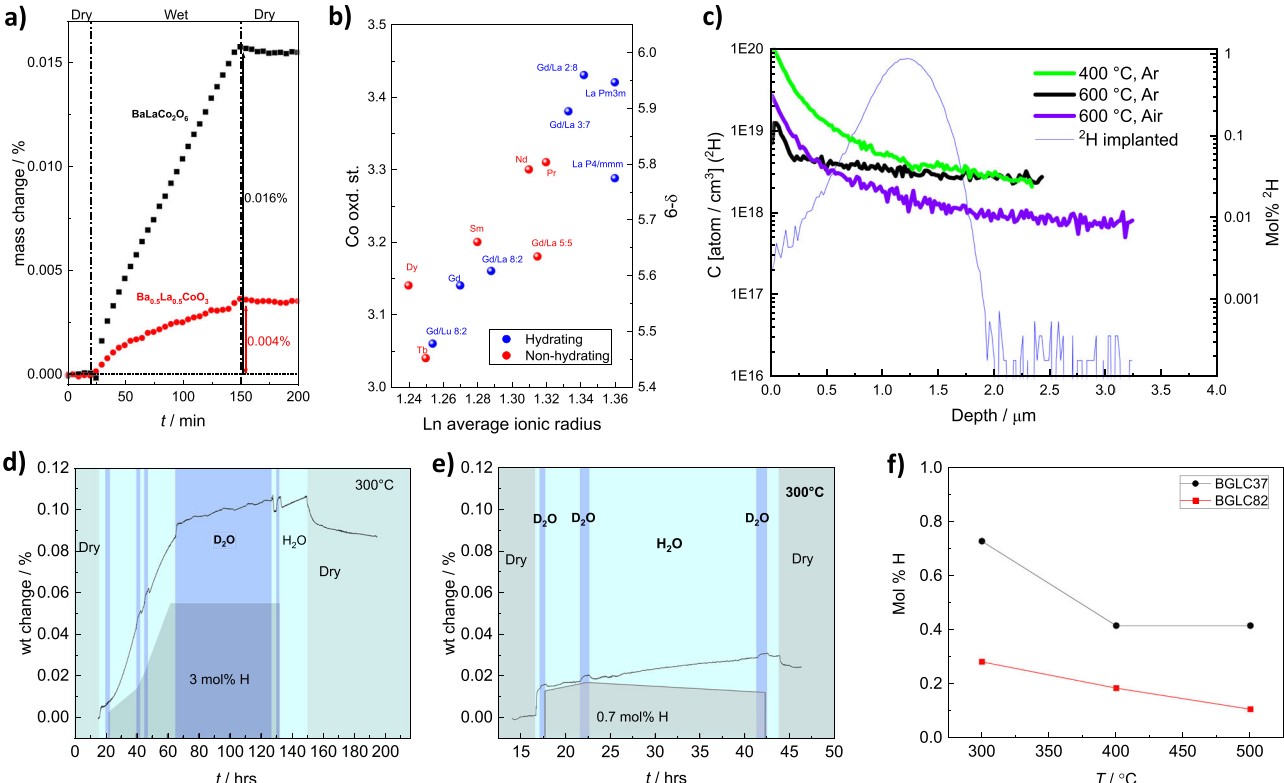

**Fig. 3 | Focused experimental investigations of hydration. a** Hydration of cubic A-site disordered $Ba_{0.5}La_{0.5}CoO_3$ versus orthorhombic A-site ordered $BaLaCo_2O_{6-\delta}$ at 300 °C in air (data adapted from ref. 20). **b** Average Co oxidation state (left y-axis) and oxygen non-stoichiometry (right y-axis) for a range of hydrating (blue) and non-hydrating (red) $BaLnCo_2O_{6-\delta}$ compositions. See Supplementary Table 1 for references. **c** $^2$H concentration profiles taken by secondary Ion Mass Spectrometry in BGLC82 after 2 h with 2.3% $D_2O$ in Ar after hydration with 2.6% $H_2O$ in air or Ar for 48 h at 600 °C or 4 h in Ar at 400 °C. Blue line in (**c**) is the concentration profile for ion-implanted $^2$H, used to calibrate absolute concentrations of $^2$H. Non-calibrated concentration profiles of $^2$H in BGLC82 and BGLC37$_{red}$ are given in Supplementary Fig. 3a, b. **d** Hydration of BGLC37$_{red}$ in wet air at 300 °C. **e** Hydration of as-prepared BGLC37 at 300 °C in air. Grey areas in (**d**) and (**e**) represent dry air, light blue represents $H_2O$ in air, and dark blue represents $D_2O$ in air. Shaded areas under the weight curves represent the hydration level calculated from $H_2O$-$D_2O$ exchanges. **f** Proton concentrations after hydration of BGLC82 and BGLC37 in air from 300 to 500 °C. Source data are provided as a Source Data file.

BLC(O) with an average oxidation state +3.3 depending on thermal and atmospheric history[20]. Figure 3a shows how hydration depends on the A-site ordering, where the ordered – and more reduced – orthorhombic BLC(O) exhibits four times higher mass gain in a wet atmosphere than the disordered – and more oxidised – cubic BLC(C)[20].

In order to further characterise how proton stability is connected to bond angles and cation-anion distance[13–15], we employed SR-PXD and NPD to evaluate differences between open- and closed-shell compositions and locate preferable proton sites. There is a slight tendency to longer TM-O bonds and a higher O-TM-O angle deviation from 180° for the closed-shell elements, especially for O4 and O6 (Fig. 1 and Supplementary Fig. 1). Taking the previous studies into account, we suggest that there is less negative charge transfer and higher proton stability at O4 and O6 in the orthorhombic *Pmmm* structure.

Figure 3b displays Co oxidation state (y1-axis) and oxygen non-stoichiometry (y2-axis) vs. average A$^{II}$-site *Ln* radius. It shows hydrating (blue) and non-hydrating (red) compositions distributed over the range of average *Ln* ionic radii, underlining the pivotal role of closed *Ln* 4f-shell over the roles of non-stoichiometry, *Ln* basicity, and average Co oxidation state.

Figure 3d, e compares hydration at 300 °C in wet air for BGLC37 after reductive annealing in Ar at 1000 °C for 10 h and subsequent equilibration in dry air at 350 °C (BGLC37$_{red}$) and for as-prepared BGLC37. X-ray diffraction (XRD) (Supplementary Fig. 2) shows that A-site ordering is more complete in BGLC37$_{red}$. As for BLC and $Ba_{0.85}La_{0.15}FeO_{3-\delta}$[16], reduced BGLC37 shows significantly more hydration than oxidised.

## D$_2$O isotope exchange; quantification of protonation

The proton concentration was estimated by fast and reversible isothermal $H_2O$-$D_2O$ exchanges in order to separate protonation and hydration from additional oxidation or reduction (Fig. 3d, e). The interpretation of the increased weight from such isotope shifts as proton concentration is associated with several potential sources of error (buoyancy, lower vapour pressures of $D_2O$ vs $H_2O$, different solubilities of humidity-reducing wetting stage salts (here KBr), and different hydration thermodynamics). We still qualitatively interpret the results to show that approximately half of the weight gain after long-term hydration in wet air is associated with oxidation.

Comparing proton concentrations in BGLC82 and BGLC37 at 300 °C in air by $H_2O$-$D_2O$ exchange (Fig. 3f), it can be inferred that even though BGLC82 has the lower Co oxidation state, the higher La/Gd ratio in BGLC37 leaves the latter with more protons, displaying dominance of *Ln* basicity over Co oxidation state.

To further quantify hydration in BGLC82 and BGLC37, $^2$H (D) concentration profiles were studied by exposing polished sintered pellets to $D_2O$ (0.023 atm) and performing Secondary Ion Mass Spectrometry (SIMS). Quantification is done for BGLC82 by calibrating $^2$H profiles using an ion-implanted reference sample, ensuring less than ±10% relative error[29]. The results are shown in Fig. 3c for samples exposed at 400 °C in Ar and at 600 °C in Ar and air. Raw counts/s for BGLC82 are compared to BGLC37$_{red}$ in Supplementary Fig. 3a, b. As expected, BGLC37$_{red}$ has orders of magnitude more $^2$H in the hydrated than in the dry state, confirmed also by isotope back-exchange in $H_2O$ atmosphere (Supplementary Fig. 3b). Raw counts/s for BGLC37$_{red}$ are 50 times higher than in BGLC82 at 600 °C in air, reflecting the higher

proton concentration in the former. The results show significant protonation of BGLC bulk up to 600 °C. For BGLC37$_{red}$, the concentration of $^2$H is uniform throughout the sub-surface and bulk, while BGLC82 shows a concentration gradient through the sub-surface. The fast back-exchange from $^2$H to $^1$H for BGLC37$_{red}$ (blue line in Supplementary Fig. 3b) indicates fast proton transport and that the concentration gradients seen in Fig. 3c represent equilibrium at each condition.

### Parallel hydration and oxidation; two processes with different time constants

Hydration of MPEC materials in wet atmospheres is often measured in long-term isothermal[5] or isobaric[1,18] TG experiments. Depending on the method and on the relative stability of oxygen vacancies, electron holes, and protons, mass exchange of water, hydrogen, and oxygen may be indiscriminately interpreted as water, as is customary for redox-stable materials, where the oxygen vacancy concentration is fixed by a dopant and electronic defects are in the minority. We have assessed this by comparing isobaric and isothermal mass gain of BGLC82 in wet air and comparing proton concentrations calculated under the assumption that mass gain in wet atmosphere stems from hydration alone in both cases (Fig. 4a, b). This approach results in a significantly overestimated proton concentration after isobaric measurements. This is due to additional – and unaccounted – oxidation upon long-term exposure to wet air during isobaric measurements. $H_2O$-$D_2O$ exchange reveals fast initial hydration coupled with slow additional oxidation, and short isothermal exposure to $H_2O$ is therefore necessary for proper determination of proton concentration from hydration under oxidising conditions.

The protonation and redox equilibria were thus further investigated for BGLC82 and BGLC37 by switches between dry and wet air at 300, 400 and 500 °C and with $H_2O$-$D_2O$ switches during equilibration. The results showed that both compositions exhibit a combination of hydration and subsequent oxidation upon exposure to water (Supplementary Fig. 4). By returning to dry conditions, new equilibria were established, indicating a structural change after exposure to water. Proton incorporation in BGLC37$_{red}$ was also examined using dry air and $N_2$ with $H_2O$-$D_2O$ switches[30]. In oxidising conditions, hydration is coupled with slow oxidation, giving rise to mass gain over 150 h under wet conditions. $H_2O$-$D_2O$ switches showed a proton concentration of 3 mol% for BGLC37$_{red}$ in wet air at 300 °C. Proton concentration was then measured after dwelling for 400 h in dry $N_2$. Isothermal switches between dry $N_2$, $H_2O$, and $D_2O$ showed that 75% of the weight gain originates from hydrogenation (Fig. 4c, d). Measuring proton concentration with increasing $pH_2O$ and $pD_2O$ (Fig. 4e) revealed an increase in the fractional hydration with higher $pH_2O$, resulting in a proton concentration of 1.5 mol% at $pH_2O = 0.016$ atm in $N_2$ at 300 °C (Fig. 4f). Hydrogenation has previously also been shown for BGLC82 substituted with 10% Ti for Co on the B-site[31].

### A-site order-disorder; atomic resolution STEM and EDS

On the basis of the acquired understanding, we compared as-prepared and hydrated BGLC37 by STEM-EDS with atomic mapping and compared the results to SR-PXD and NPD to unify local and global characterisation of A-site order-disorder upon hydration and oxidation. Figure 5a shows atomic maps of BGLC37 in dry and hydrated states, verifying the A-site ordering of Ba-*Ln* in the dry state and A-site disorder after hydration. Ba is ordered on the A$^I$-site with a minor occupancy also on A$^{II}$ in the dry state (Supplementary Fig. 5). Hydration induces the A-site disordering, i.e., BGLC37 shows a transition from A-site order to disorder after exposure to a humid atmosphere at 300 °C. Around 30 grains were examined without detecting A-site order in the hydrated state. SR-PXD Bragg reflections are, however, indistinguishable between dry and wet states, both for BGLC37 and BGLC82[19], but may also be refined with a *Pmmm* 2a3b3c supercell

(Supplementary Fig. 6a, b), which is oxidised and fully A-site disordered. The combination of SR-PXD, NPD (Supplementary Fig. 7), and STEM-EDS mapping reveals a transition from reduced, A-site ordered *Pmmm* a2b2c for BGLC37 and *P4/mmm* ab2c for BGLC82 in the dry state, into an oxidised, A-site disordered *Pmmm* 2a3b3c supercell for both compositions upon hydration (Fig. 5b). Oxygen vacancies in the *Ln* layer are suggested to compensate for the difference in ionic radii between the cations occupying the A$^I$ and A$^{II}$ sites[24]. Hence, large *Ln*'s such as La will oxidise and disorder more easily due to the smaller difference in ionic radius between Ba and La. For BGLuC82, the two lanthanides are sufficiently small to maintain the same oxidation equilibrium before and after hydration. This is reflected by the full reversibility when reverting from wet to dry conditions (Fig. 2a).

The hydrated supercell is oxidised, but also contains oxygen sites such as O10 that – due to higher bond angle deviation from 180° and elongated Co-O bond[13–15] – will have a more complete Co-O electron transfer and can be protonated more easily than O4 and O6 in the a2b2c phase. As a result, both hydration and oxidation become more favourable when the structure changes from a2b2c to 2a3b3c, and hydration – a process consuming oxygen vacancies (reaction A in Fig. 5c) – induces A-site disorder and oxidation (Reaction B in Fig. 5c) in a combined A + B hydroxidation reaction.

Under inert conditions, exposure to a wet atmosphere is not enough to trigger the A-site disorder. This is most likely due to the larger concentration of oxygen vacancies in the Ln layer stabilising the a2b2c structure at low $pO_2$. Instead of A-site disorder and oxidation, hydration is combined with reduction in an A-B hydrogenation reaction (Fig. 4e, f).

## Discussion

We have seen that hydration, according to

$$H_2O(g) + v_O^{\cdot\cdot} + O_O^x \rightarrow 2OH_O^{\cdot} \qquad (1)$$

in p-type positrode materials may be accompanied by oxidation under oxidising conditions:

$$\frac{1}{2}O_2(g) + v_O^{\cdot\cdot} \rightarrow O_O^x + 2h^{\cdot} \qquad (2)$$

This may arise because the material was in a metastable or frozen-in reduced state (e.g., after high-temperature annealing) or because the hydration leads to dimensional (bond length and bond angle) changes in the crystal structure and, in turn, the electronic structure that governs oxidation. In some cases, these changes promote slow A-site disorder, which itself increases hydration and oxidation. The weight of oxygen from oxidation may be misinterpreted as hydration in TG studies, and the multiple entangled processes may proceed at different rates, making interpretation complicated. Under less oxidising conditions, the reverse of Eq. 2 (reduction) may occur, and by subtracting Eq. (2) from Eq. (1), we get what we may refer to as hydrogenation:

$$H_2O(g) + 2h^{\cdot} + 2O_O^x \rightarrow 2OH_O^{\cdot} + \frac{1}{2}O_2(g) \qquad (3)$$

The weight uptake from this process is only due to the protons. As shown in this work, $H_2O$-$D_2O$ isotope exchanges are helpful to properly quantify the concentration of protons and correctly interpret hydration in correct contributions from Eqs. 1–3, as well as understanding the underpinning slow structural changes.

The *Ln* 4f0, f7 and f14 effect on hydration is reflected in the electron structure through the depopulation of the Co3d $t_{2g}$-O2p bond (Fig. 2c–h). This correlation can be supported by recent reports of hybridisation between *Ln* 4f and TM3d orbitals[32].

To summarise, 45 double perovskite compositions with the general formula $A^I A^{II} B_2 O_{6-\delta}$, having Ba or Ba+Sr on $A^I$, a mix of lanthanides $Ln$ on $A^{II}$, and Co on B, have been studied to establish the general

principles that govern hydration. Two of the three compositions with the highest values in the hydration screening (Fig. 2b) – BLC(O) and BGC – are considered structurally unstable under PCEC operational

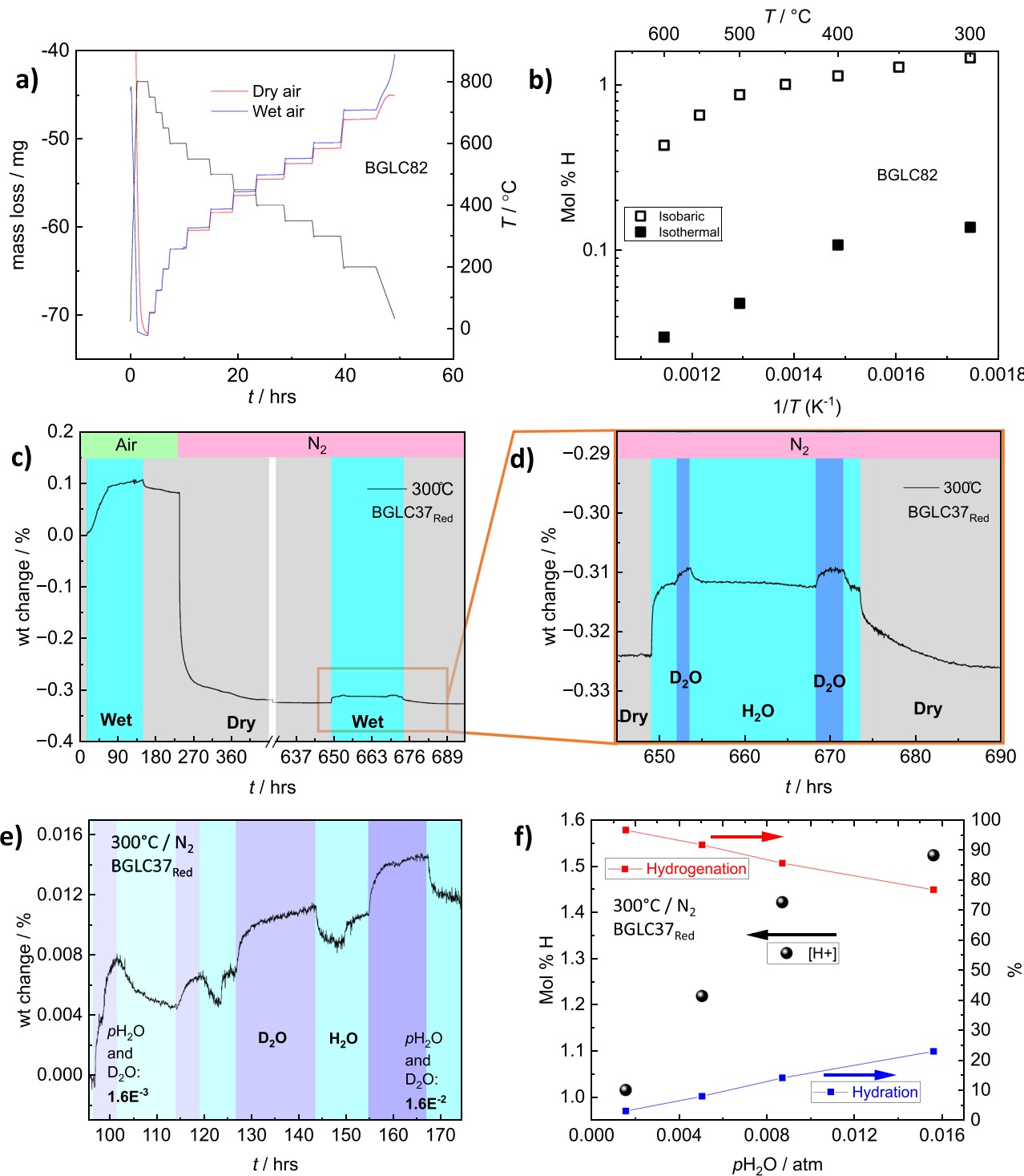

**Fig. 4 | Thermogravimetry of hydration, isotope exchange, and hydrogenation. a** Isobaric TG measurements of BGLC82 in 50 °C temperature steps from 800 to 200 °C in dry and wet conditions. **b** Hydration of BGLC82 calculated from isobaric and isothermal weight gain in wet air versus dry. **c** TG curve of BGLC37$_{red}$ in dry (grey background under green header) and wet (light blue background under green header) air, long-term dwell in dry $N_2$ (grey background under light red header) and during hydration in $N_2$ (light blue background under light red header). **d** TG curve showing enlarged part of (**c**) with mass gain in BGLC37$_{red}$ in dry (grey background)

and wet $N_2$ with $H_2O$ (light blue)-$D_2O$ (dark blue) switches at 300 °C. The mass gain difference between $H_2O$ and $D_2O$ enables the exact determination of the proton concentration in wet conditions and thus also the determination of fractional hydration versus hydrogenation. **e** TG curve with mass gain at 300 °C in increasing $pH_2O$ (lighter blue) and $D_2O$ (darker blue) switch at all $pH_2O$ in $N_2$ for BGLC37$_{red}$. **f** Proton concentration (left axis) and fractional hydration and hydrogenation (right axis) in BGLC37$_{red}$ with increasing $pH_2O$ in $N_2$. Source data are provided as a Source Data file.

conditions, and they are used as examples here, more than as potential triple conducting positrode materials. XAS measurements confirm lower hole concentrations in the Co3d $t_{2g}$-O2p bond for hydrating compositions, and that proton-hole repulsion induces a shift in electron occupancy from $t_{2g}$ to $e_g$. Introducing intermediate lanthanides or substituting Sr for Ba diminishes hydration. The B-O bond length and – angle for a range of $Ln$ –compositions have been investigated (Supplementary Fig. 1), showing some correlation between the

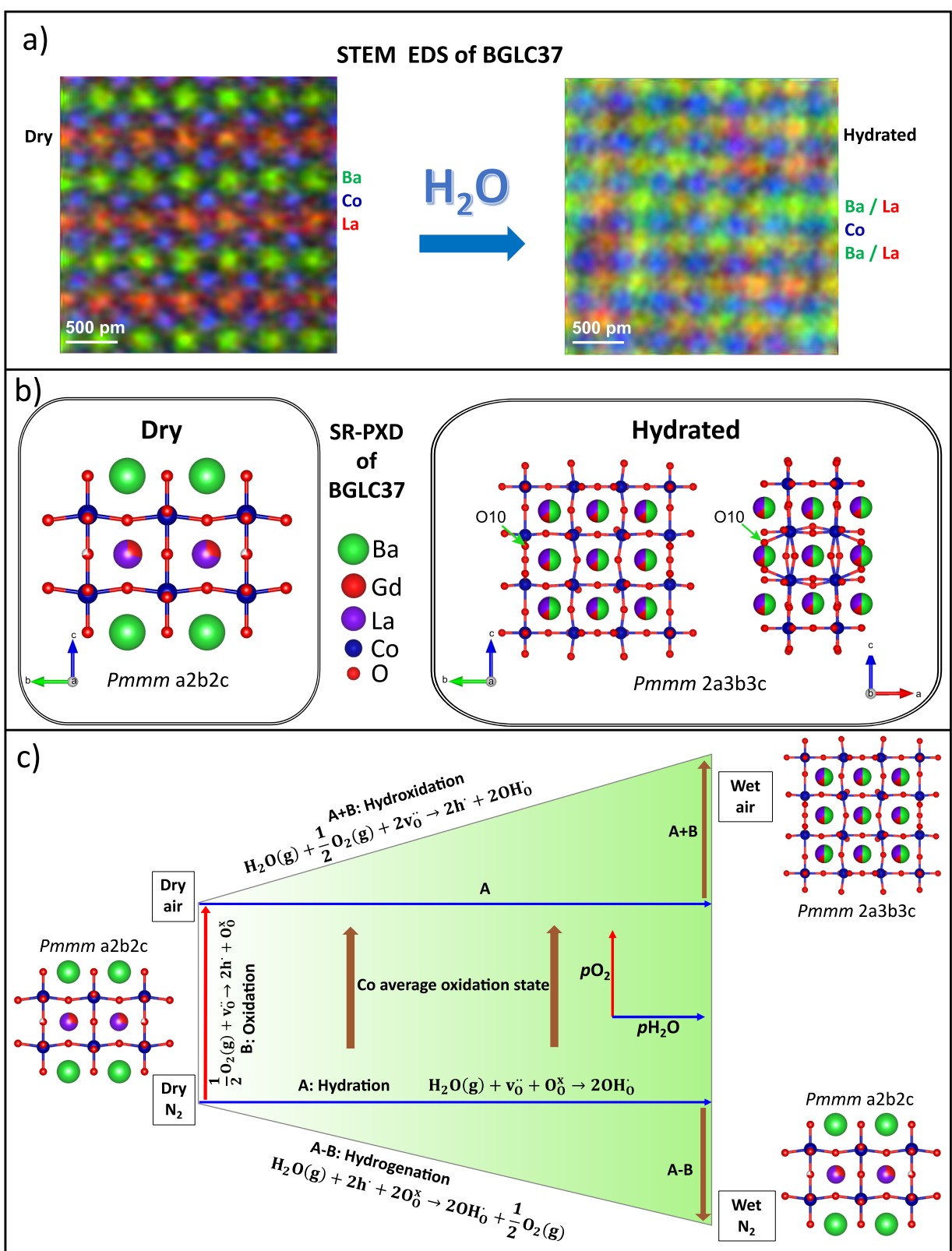

**Fig. 5 | Structure-hydration-redox relationships. a** STEM-EDS maps of Ba, La and Co in dry, ordered (left) and hydrated, A-site disordered (right) state. **b** Structures of BGLC37 in dry state refined to *Pmmm* a2b2c (left) and in wet state refined to *Pmmm* 2a3b3c (right). **c** Reaction schematic for hydration (A), oxidation (B), hydrogenation (A-B) and hydroxidation (A + B), with accompanying structural polymorphs for dry and wet conditions.

average Co-O-Co bond angle in the O4 and O6 position and hydrating compositions. Hydration is shown to be higher for more basic Ln compositions (Fig. 3f) and to increase with lower cobalt oxidation state (Fig. 3a, d, and e), with basicity being dominant over Co oxidation state (Fig. 3b, f). Proton concentrations of 0.4 and 0.9 mol% in BGLC82 and BGLC37, respectively, have been determined by $H_2O$-$D_2O$ exchange under wet oxidising conditions. Bulk hydration is confirmed by $^2H$ concentration gradients obtained by SIMS after $^1H$-$^2H$ isotope exchange. The hydration process is shown to be a combination of hydrogenation and oxidation, depending on $pO_2$. STEM-EDS, SR-PXD, and NPD unambiguously show that hydration triggers a transition into an oxidised A-site disordered 2a3b3c *Pmmm* supercell with favourable proton positions and that this A-site order-disorder alters both oxidation and hydration equilibria.

A general overview of the hydration, oxidation, and hydrogenation reactions and how they depend on the atmosphere is given in Fig. 5c. The different reactions can be expressed as combinations of hydration (A, x-axis) and oxidation (B, y-axis). The refined structures of BGLC37 illustrate the combined effects of $pO_2$ and $pH_2O$ and the tendency of A-site disorder. We note that pure hydration – without accompanying redox – in mixed valence oxides like the ones we consider here occurs only under specific combinations of $pO_2$ and $pH_2O$. In this class of materials, redox stability is connected to the structural stability of the A-site ordered polymorph, mostly depending on the difference in ionic radii between the cations occupying the $A^I$ and $A^{II}$ sites.

Beyond the above relationships, this work's most important finding and hence governing principle of hydration of double perovskites is that all hydratable compositions contain lanthanides with empty, half-full, or full 4f electron shells, applying to La, Gd, and Lu, respectively. The consequence of this finding is an opportunity to adjust the population of electrons and holes – and thus the charge transfer and ionicity – of the TM-O bond through the A-site cation. The electronic environment in the TM-O bond is detrimental for proton stability and redox activity, and this governing principle hence unlocks a tool for the design of mixed proton-electron positrodes maximising hydration and surface reactions kinetics.

## Methods

### Synthesis

$Ba_{1-x}Sr_xLn_{1-x}Ln'_xCo_2O_{6-\delta}$ ceramics (*Ln*, *Ln'* – lanthanides or yttrium, $0 \geq x \geq 1$) were synthesised using the solid state reaction method according to the protocol established in our previous work[20]: $La_2O_3$ (99.99% Alfa Aesar), $Pr_6O_{11}$ (99.99% Aldrich) and $Nd_2O_3$ (99.9%, Chempur) were preheated at 900 °C for 5 h. Additionally, $Sm_2O_3$ (99.9%, Chempur), $Gd_2O_3$ (99.98% Alfa Aesar), $Tb_4O_7$ (99.99% Aldrich), $Dy_2O_3$ (99.9%, Chempur) and $Y_2O_3$ (99.9%, Chempur) were used as the *Ln/Ln'* sources. The *Ln/Ln'* oxides, $BaCO_3$ (99.9% Sigma Aldrich) and $Co_3O_4$ (99.98% Alfa Aesar), were mixed in an agate mortar in stoichiometric proportions and then pelletized. The synthesis was performed in a muffle furnace at 1150 °C for 48 h. To form the tetragonal $BaLaCo_2O_{6-\delta}$, the as-synthesised sample was heat-treated in Ar at 1050 °C for 24 h and then re-oxidised in air at 350 °C for 3 h. BGLC37 and BGLC82 were commercial (Marion Technology, France).

### Diffraction

SR-PXD patterns were obtained at the Swiss-Norwegian Beamline (SNBL) BM01 and BM31, ESRF, Grenoble, with a Pilatus 2M 2-dimensional (2D) detector (BM01) and Dexela Perkin Elmer 2923 CMOS 2D detector (BM31) with wavelengths of 0.78956 Å and 0.31232 Å, respectively. 1D data were obtained by integration of the 2D diffraction patterns using the Bubble programme. Additional measurements were performed at the Diamond Light Source in Didcot, UK, at beamline I11 in high-resolution mode by 45 photomultiplier

detectors, each with a Si(111) analyser crystal ($\lambda = 0.82657$ Å), cf. an earlier publication for details[20].

X-ray diffraction (XRD) was made with an X'Pert Phillips diffractometer with Cu Kα radiation (1.541 Å).

NPD data were collected at RT using the constant-wavelength diffractometer PUS ($\lambda = 1.555$ Å) at the JEEP II reactor (Kjeller, Norway) and the time-of-flight (TOF-NPD) instrument POWGEN at SNS (Oak Ridge, US). The sample was loaded in vanadium cans, and results analysed using the Rietveld method, refined simultaneously to the SR-PXD and NPD data, when available, see an earlier publication for further methodological details[20] and refinement parameters in Supplementary Methods.

### Thermogravimetry

Thermogravimetric hydration measurements were performed at the University of Oslo (UiO) and at Gdańsk University of Technology (GUT). At UiO, a Netzsch STA 449 F3 Jupiter was coupled to a custom-built gas mixer[33] equipped with $H_2O$ and $D_2O$ wetting stages, where dry $N_2$ was humidified with 2.6% $H_2O$ or 2.3% $D_2O$. In order to keep $pO_2$ constant in dry and wet atmospheres, the humidified $N_2$ was mixed with dry $O_2$ and $N_2$ to reach a final composition of 1.6% $H_2O$ (or 1.4% $D_2O$) in air. At GUT, a Netzsch Jupiter® 449 F1 simultaneous thermal analyser was used for water uptake studies isothermally at 300 °C. As-prepared powders, crushed from pellets, were heated at a rate of 5 °C/min to 800 °C and kept for 5 h to remove the residual water. Then the sample was cooled to 300 °C at the same rate and stabilised for 2 h before the purge gas atmosphere was switched from dry to humidified (1.9% $H_2O$). This was maintained for 2 h, followed by the reverse switch to dry gas. Synthetic air was used as a purge gas, and $N_2$ as a protective gas. $H_2O$-$D_2O$ exchanges were performed for selected compositions to support the estimation of proton concentrations based on hydration as the initial dominating process during equilibration in wet conditions. The isobaric hydration experiment presented in Fig. 4b was performed by heating the powder in dry air from room temperature to 900 °C, dwelling for 1 h, and stepwise cooling at 3°/min and equilibrating at each step temperature. The process was then repeated in a wet atmosphere, and the weight difference between wet and dry atmosphere at each temperature was interpreted as pure hydration. See also the Supplementary Discussion of Hydration of a secondary phase.

### SIMS

Four sintered BGLC82 pellets sintered at 1190 °C for 10 h for sufficient relative density were used for the SIMS experiments. Two samples were equilibrated for 48 h at 600 in 2.6% $H_2O$ in air and Ar, respectively, before switching to 2.3% $D_2O$ for 2 h. A third sample was equilibrated at 400 °C and 2.6% $H_2O$ in Ar for 48 h before switching to 2.3% $D_2O$ for 4 h. A fourth sample was annealed in dry air at 600 °C for 48 h and used for calibration by ion implantation. For BGLC37$_{red}$, the same powder was used as had been used during the long-term hydration shown in Fig. 3d. The powder was pressed into pellets and sintered in ambient air for 10 h at 1190 °C. The pellets were polished and annealed in dry air at 900 °C for 6 h and then for 53 h in dry air at 600 °C before being cooled to room temperature. Following this treatment, one pellet was heated to 600 °C in a TG in dry air and equilibrated for 15 h before introducing 2.3% $D_2O$ for 280 minutes. A second pellet was exposed to the same treatment, now keeping the pellet for 50 h in 2.3% $D_2O$. A third pellet was kept in its dry state. A Cameca IMS7f SIMS equipped with a 15 keV $O_2^+$ primary ion beam source was used to record the concentration vs. depth profiles of $^2H$. The raster size was set to 150 µm, and data was collected from the central 33 µm. Absolute concentration of $^2H$ was obtained by measuring a separate $^2H$ implanted reference sample.

The SIMS reference standard was prepared by ion implantation using a 1 MeV Tandem ion accelerator from National Electrostatics

Corporation (NEC). Deuterium ions were extracted from a gas source and accelerated to an energy of 200 keV and implanted into a separate sample to a fluence of $5 \times 10^{14}$ cm$^{-2}$. The sample was used as a reference sample to convert the measured deuterium signal into absolute concentration. The SIMS reference sample was measured with the same instrument parameters as the sample of interest, utilising the implantation fluence to calculate the conversion factor, i.e. the relative sensitivity factor of deuterium.

For depth calibration, the sputtered crater depths were determined by a Dektak 8 stylus profilometer and a constant erosion rate was assumed. After having obtained $^2$H concentration profiles by SIMS, one deuterated pellet was heated in a TG to 600 °C in 2.3% $D_2O$ in air, equilibrated at 600 °C for 15 min before switching to 2.6% $H_2O$ for 20 min. The $^2$H profile was subsequently collected after this reversed isotope exchange. All pellets were rapidly cooled to room temperature after exposure to $D_2O$.

## XAS

X-ray Absorption Spectroscopy was performed at the Solaris National Synchrotron Radiation Centre in Kraków, Poland. A dedicated PEEM/XAS bending magnet beamline was utilised to measure O-K edges. Powder samples were mounted on carbon tape and placed on Omicron plates. The measurements were performed under ultra-high vacuum at room temperature. The presented data were collected in the XAS fluorescence mode. The data were normalised according to the conventional procedure to the edge intensity after subtracting the background signal, and fitted with a first-order polynomial fitted to the pre-edge regime. Before the measurement, the samples were dried for 24 h at 900 °C in air to obtain the 'dry' state. Samples labelled as 'wet' were exposed to humidified air for 24 h at 300 °C.

**STEM/EDS.** The microstructure and composition of the samples were characterised by annular bright field (ABF) and high-angle annular dark field (HAADF) scanning transmission electron microscopy (STEM) coupled to X-ray energy dispersive spectroscopy (EDS). This work was performed with a DCOR Cs probe-corrected FEI Titan G2 60-300 instrument, with 0.08 nm nominal spatial resolution when operated at 300 kV, equipped with a Bruker SuperX EDS system comprising four silicon drift detectors. Sample preparation involved depositing powder suspended in isopropanol on lacey carbon copper grids, followed by shielded plasma cleaning.

## Iodometric titration

Iodometric titration was used to determine the oxygen stoichiometry and average cobalt oxidation state at room temperature, performed as described in previous works[20,34]: 15–20 mg of the sample and a surplus of potassium iodide (ca. 0.2 g) were put in a three-neck flask, flushed with nitrogen. When the air was removed, 15 ml 2 M HCl was added to dissolve the sample, and iodine ions were titrated in an inert atmosphere with 0.01 mol/dm$^3$ $Na_2S_2O_3$ from a 10 ml microburette (±0.05 ml) using starch as an indicator.

## Data availability

The experimental data generated in this study are provided in the Supplementary Information/Source Data file. Source data are provided with this paper.

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

## Acknowledgements

This work is financed by The Research Council of Norway (Grant nº 272797 "GoPHy MiCO" and nº 299736 "FunKeyCat") through the M-ERA.NET Joint Call 2016 and 2018 (R.S., M.H.S., T.N.). Project "FunKeyCat" is supported by the National Science Centre, Poland, under the M-ERA.NET 2, which has received funding from the European Union's Horizon 2020 research and innovation programme under grant agreement no 685451 (2016/22/Z/ST5/00691) (S.L.W., A.M.G., M.G.), and the Spanish Government through the National Research Agency AEI (grant PCIN-2017-125) (M.B., J.A.S.). This publication was partially developed under the provision of the Polish Ministry and Higher Education project "Support for research and development with the use of research infrastructure of the National Synchrotron Radiation Centre SOLARIS" under contract nr 1/SOL/2021/2 (S.L.W., A.M.G., M.G.). We acknowledge the SOLARIS Centre for access to the Beamline PIRX, where the measurements were performed. We thank Dr. Marcin Zając for assistance (proposal nos. 191011 and 201036). We also thank Dr. Vegar Øygarden for Rietveld refinements of NPD data.

## Author contributions

R.S.: Conceptualisation (lead), data curation (equal), formal analysis (equal), funding acquisition (lead), investigation (equal), methodology (equal), project administration (lead), supervision, validation (lead), visualisation (lead), writing original draft (lead). T.N.S.: Investigation – SIMS measurements on BGLC82. L.V.: Investigation – SIMS measurements on BGLC37. P.A.C.: Investigation – STEM-EDS. M.H.S.: SR-PXD and NPD, data curation (equal), formal analysis (equal), funding acquisition (equal), investigation (equal), methodology (equal). S.L.W.: Formal analysis (equal), funding acquisition (equal), investigation (equal), methodology (equal), project administration (equal), supervision, validation (equal), writing (equal). I.S.: Formal analysis (equal), investigation (equal), methodology (equal), validation (equal). A.M.G.: Formal analysis (equal), funding acquisition (equal), investigation (equal), methodology (equal), project administration (equal), supervision, validation (equal). M.B.: Formal analysis (equal), funding acquisition (equal), investigation (equal), methodology (equal), project administration (equal), supervision, validation (equal). T.N., M.G. and J.M.S.: funding acquisition (equal), investigation (equal), methodology (equal), project administration (equal), supervision, Writing – Review & Editing.

## Competing interests

The authors declare no competing interests.
