## [Transparent Peer Review file · Nature Communications]

Governing principles of hydration of mixed proton conducting Co-based double perovskites

Corresponding Author: Professor Truls Norby

Version 0:

Reviewer comments:

Reviewer #1

(Remarks to the Author)

Overall I have a positive tendency to recommend this paper for publication in Nat. Comm. Nevertheless I also suggest significant revision on various levels.

The topic is highly relevant and suitable for Nat Comm. Mixed conductors which include also proton conductivity are key functional materials for protonic ceramic fuel and electrolyzer cells, which will acquire an increasingly important role in a H₂-based energy system. Several groups find that on one hand, Co-rich materials have a tendency of less pronounced proton uptake, on the other hand Co is typically beneficial for the catalytic activity of such electrode materials. Thus, an improved understanding of a Co-based material as given here is highly welcome.

The double perovskites BGLC and related RE-substituted materials have been investigated by the present authors' groups over an extended time. The present paper finalizes these investigations and is able to draw a quite clear picture of a materials system that is really complex. For example, it is found that - even for non-redoxactive RE³⁺ cations - the amount of hydration depends on RE³⁺ being f0, f7 or f14 configuration. And while the amount of proton uptake is quite low, it can still modify the ordering of the cations and propensity for oxidation, which leads to relaxation processes after pH₂O step on multiple time scales.

Therefore the paper is recommended for publication, but in order to be well comprehensible for the readers the presentation of this complicated story needs some improvement.

Presentation:

(1) the paper covers a large range of samples, several data measured also in preceding publications. It would be extremely helpful to collect all lattice parameters, other structural information (space group, presence/absence of A site order etc, quality of the Rietveld fitting) in a table in the SI, indicating also exact state of sample (as-prepared, dry annealing, O₂ or N₂ annealing, hydrated state...)

(2) the paper shows that TG weight change may originate from hydration, hydrogenation, oxidation, and how the actual proton content can be determined from H₂O/D₂O exchange. Table S1 states "hydration mol% from weight gain in H₂O" - so it seems the numbers there might over/underestimate the real proton concentration? Please add a column giving the real [H⁺] as determined from H₂O/D₂O. Labelling in fig 2b is misleading: axis label says "mol% H" which sounds like real proton conc. (from H₂O/D₂O) while cation says "weight gain in H₂O" which is sum of all processes. BGLC82 value in fig 2b differs from that in fig 4b

(3) fig 2b is quite puzzling with the colored arrows. better use just red symbols for "hydrating" and black for "non-hydrating" samples. Fig 3d,e: giving absolute weight changes in mg is not very suitable (magnitude depends on absolute sample weight used). I understand that not always the precise [H⁺] is known owing to different reactions, but at least use "weight change/%" instead of "weight change/mg". Generally, the colored backgrounds in the TG plots (also fig 4) are too strong. Please use lighter colors, and try to simplify the figures as much as possible (i.e. not using color gradients when within such a field the conditions are not changing)

(4) fig 4: it would be helpful to insert the sample name abbreviation in each of the plots. Fig 4ab please discuss in more detail in the text that/why the isothermal process (does it proceed by hydration or by hydrogenation, which reaction is assumed for converting weight change into proton conc.) yields lower proton conc. than isobaric (does this imply the isobaric value is overestimated). Fig 4cd please indicate T. It would be nice to directly indicate in this fig. which fraction of H is from hydration and which from hydrogenation. Fig. 4e please indicate T, that it is in N₂, and what are the pH₂O and pD₂O.

(5) fig 5 middle part: please write "dry" and "hydrated" close to the respective structures. Bottom part: the small coordinate system with axes labelled pO₂ and pH₂O is easily overlooked. Please indicate more prominently that horizontal axis refers to proton concentration, and vertical axis to oxygen uptake (oxidation). Please indicate more clearly that the "hydroxidation"

happens only because the hydration leads to A site disorder which then favors further oxidation (without this it is not conceivable why hydration and oxidation should occur in parallel since they both consume oxygen vacancies)

Additional discussion/explanation:

(6) within the presented data set there is a strong correlation that only f0, f7 or f14 RE3+ materials show large hydration. However, based on the usual "textbook knowledge" that f orbitals have a very small radius and are thus "quite inside the atom" it is not so obvious how the details of f occupancy (6 or 7 or 8 electrons) would affect the t2g orbitals of Co which at approx. 3.5 Angstrom distance. Might it help to think about Jahn-Teller-like effects at RE which could modify RE-O bonding, or generally how f occupancy could affect RE-O bonds (and this then the Co-O bonds)? Along this line, maybe look also at RE-O bond lengths and angles (similar to Co-O in fig A1ab)?

(7) fig 2d-h: it is impressive but to some degree also astonishing how a small proton uptake of about 1% (0.01 H per AA'B2O6 !) strongly modifies the intensity of the O-prepeak (which is averaging over all O's, most being unprotonated). Any further explanation is welcome

(8) Co is known to have several spin states. What is known about spin of Co in present samples, it may also affect nature of Co-O bond and thus tendency for protonation? maybe add some information in the SI

(9) hydration vs hydrogenation are quantified from the H2O/D2O experiment. However, even when pH2O=pD2O are used, the hydration mass action constants for H and D might slightly differ because zeropoint vibration effects. At least mention that this might add a little uncertainty

Further points:

- fig 1b please write "ion radius ratio" instead of "IRR"

- caption fig 2 there is no "g-k", should be d-h?

- fig 3b: authors seem to use a different scale for ionic radii than in [20] - why? the radii in [20] are the wellknown Shannon radii, with 1.16 Å for La in CN=8

- ion implantation is not a standard lab technique. Thus it must be specified how/here the D implantation as SIMS standard was done

- pg 9 line 225 should probably read "hydration causes A site disordering"?

- lines 263 and 267 please mention the figure numbers which show the respective correlations

- fig S6 maybe indicate more clearly which are the specific reflections that evidence the A site ordering. maybe use log scale for intensity to give more weight to small reflections

- standard sample preparation is 1150°C for 48h - what is resulting relative density. Does this directly yield sufficiently dense samples for the SIMS experiment, or were they made by modified process?

- I guess "powder" for TG mentioned in line 340 is obtained by crushing the 1150°C for 48h pellets ? then please write that. Any idea of resulting grain/particle size?

Reviewer #2

(Remarks to the Author)

This paper studies Co-based double perovskites with mixed A2+ and A3+ cations in the A-site. It employs a variety of structural, thermogravimetric and electrochemical measurements to draw general conclusions about the hydration behavior.

Although the manuscript contains some interesting techniques, I found it to be unorganized in the presentation of data and results, and this regrettably influenced clarity. In general, the discussion and conclusions are insufficiently supported by the results. In addition, the discussion frequently resorts to data that was already published, or even not shown.

I recommend that this paper is submitted to a more specialized journal after its several problematic aspects are amended. Specific points follow.

1) Most figures are assembled out of different panels, which can help condensing information, but then the meaning of the large Figure 1 is unclear.

2) The authors mention that both neutron (NPD) and synchrotron (SR-XRD) powder diffraction have been used to "unambiguously show that hydration triggers a transition into an oxidised A-site disordered" supercell (page 11). I could not find NPD in either the manuscript or the Supporting information. On the other hand, page 1 only mentions SR-XRD as the technique to refine the position of seven different oxygen sites.

3) The abstract anticipates that hydration is linked to A-site basicity, but this is actually never discussed directly.

4) One of the main hypotheses put forward is that closed or semi-closed 4f states (La3, Lu3+, Gd3+) influence hydration, and the authors use open/closed symbols in Fig 2b and Table S1. There seems to be inconsistency here: why is BGLYC721 marked as closed shell but BGLuC721 is not? Why are BGLSrC(C) or BGLuC721 not closed shell? The trend seems less clear than it is presented by the authors, as very similar compositions show markedly different hydrations.

5) Fig 2a is mostly taken from ref. 20. Fig2b is plotted against IRR as the x-axis: what is the meaning of the radius ratio and why should hydration be plotted against it? This is never explained. In Figure 3b the x-axis seems to be way too low, and I suspect the ionic radii for an octahedrally coordination were incorrectly considered.

6) As the authors underline how the extensive composition matrix (page 2) supports the correlations, one would expect that the XAS spectra on Fig 2c-h would cover the same samples as in Fig 2b, while it only Gd/La compositions are shown. Also since the absolute intensity of XAS peaks is discussed, some more information about measurements (is it electron yield? Are surface states expected to be the same as the bulk?) and data reduction (how are the spectra normalized?) would be good. Also panels c-e and f-h are shown with a common x-axis, but the tickmarks don't align.

7) At least laboratory XRD with Rietveld refinements should be shown for each sample in the Supporting information. The only diffraction data actually shown is Fig S6, which is too small to appreciate any difference between the two panels. There is also no goodness of fit or statistical significance reported in order to tell the two models apart. Previous data by the same authors (refs. 20 and 28) is much better presented in comparison.

Reviewer #3

(Remarks to the Author)

This study presents the governing principles of protonation by systematically analyzing the hydration trends of cobalt-based double perovskite, $A^I A^{II} Co_2 O_{(6-\delta)}$. The authors report that lanthanide A-site dopants with a closed or semi-closed Ln 4f shell (i.e., Ln = La, Gd, Lu) induce electron hole transfer from eg to t_{2g} in the Co-O bond, as supported by XAS measurements. Additionally, STEM-EDS, SR-PXD, and PND clearly demonstrate that hydration triggers a transition to an oxidized A-site disordered 2a3b3c Pmmm supercell, which affects both oxidation and hydration equilibria. This work is well-designed, and the analysis is meaningful, particularly in designing mixed proton-electron positrodes that maximize hydration and surface reaction kinetics. Such advances could offer substantial contribution to the PCEC community. Therefore, I would recommend that this paper is suitable to be published in this journal after revision with following comments.

- In Figure S3, the concentration profile of BGLC37red with D₂O exchanged at 600°C shows a distinct pattern of initially increasing, then decreasing, and finally reaching saturation. This behavior is not mirrored in BGLC82. Could the authors elaborate on the underlying reasons for this unique trend in BGLC37red?

- The authors state that the hydrated supercell is oxidised and contains oxygen sites like O10, which are more easily protonated than O4 and O6 in the a2b2c phase, leading to enhanced hydration and oxidation in the 2a3b3c structure. Could the authors provide further explanation or references to support this assertion? Understanding the specific reasons why O10 sites are more susceptible to hydration compared to O4 and O6 would clarify this aspect of the study.

- Regarding a STEM-EDS results as provided in Figure 5a, the authors mentioned that the ordering structure in BGLC37 become disorder after hydration. I'm simply wondering if this phenomenon may be reversed. Does hydrated BGLC37 back into an ordered structure after dehydration? Furthermore, do you have a change to check the chemical stability under hydration conditions?

- In figure 2a, I think there are some missing TGA results. Please show all TGA data with all compositions even in supplementary figure.

- Why the mass change of BLC with cubic structure and BLC with ordered structure is more than 4 times different. Is it because the unit cell volume has increased? Please clearly explain this point.

- Based on the TGA data in Figure 2a, fully Gd and La doped in BaLnCo₂O_{6-d} samples seem to exhibit the highest mass change under both dry and wet conditions. Nonetheless, Gd and La co-doped sample, BGLC, is used as representative material in this work. There is a lack of clear explanation regarding this.

Minor points:

- There are errors in Figure 2. In caption b), "See also Table S2 " need to be changed "See also Table S1". In figure 2b, there is duplicate data point with an IRR close to 1.15 (BSrPC). Please check it.

- There are typos on page 6. In figure 3b, y1-axis is Co oxidation state and y2-axis is oxygen non-stoichiometry. And blue arrows indicate hydrating and red arrows indicate non-hydrating in figure 3b. It seems to be incorrectly labeled, unlike the manuscript. Please revise these errors.

Reviewer #4

(Remarks to the Author)

This work presents a new governing principle for the hydration and proton transport in Co-based double perovskites. This principle is extracted from a comprehensive experimental campaign that includes the preparation of a large material library (experimental matrix based on Ln combinations) and fundamental characterization looking at the atomic-level structural and electronic aspects together with proton-transport methods.

The quality and completeness of the presented results are of the highest quality. The work is generally well described, and the discussion is well articulated and succinct. I am convinced that this article will impact the research community working on proton ceramics and electrochemistry. I recommend the publication of the paper subjected to minor revision, taking into account the following list of comments and suggestions:

1- Fig. 2b. This chart is difficult to read, and I needed to go to Table S2 (back and forward) a couple of times. Is there a simple way to include the composition information directly in Fig. 2.

2- Page 6. When comparing proton concentrations in BGLC82 and BGLC37, it is properly concluded that there is a dominance of Ln basicity over Co oxidation state. For the general reader, I recommend the authors explain more in detail the term 'Ln basicity' in this context and its implications in the proton and electronic transport.

3- Last paragraph, page 6. Please, use a unique symbol for deuterium. Now, D and 2H are used simultaneously in the text, Fig. 3 and Fig. S3

4- Page 9. There is a very interesting conclusion: "the two lanthanides are sufficiently small to maintain the same oxidation equilibrium". Please, could you elaborate more on this and extrapolate to other combinations and systems, i.e., as a general principle?

5- Fig. 5. I suggest the authors label each individual image and re-arrange it to improve the overall readability of the figure. In addition, as a general conclusion of the figure, it is stated, "We note that pure hydration in mixed valence oxides like the ones we consider here occurs only under specific combinations of pO₂ and pH₂O." Can the authors be a bit more specific on the type of combinations pO₂-pH₂O enabled hydration and the associated materials properties that determine this 'hydration space'.

6- To illustrate the potential and versatility of the proposed principles, the authors could comment on the potential application of these principles to the design and understanding of other mixed oxides exhibiting hydration and/or proton transport, if possible, providing specific examples of composition and crystalline structures. Can this be connected with disorder or entropic effects in different classes of oxides?

Version 1:

Reviewer comments:

Reviewer #1

(Remarks to the Author)

the revision has sufficiently resolved my questions, now it is recommended for publication

Reviewer #2

(Remarks to the Author)

In the response letter and revised paper, the authors have partially addressed my earlier comments. However, my general assessment of this paper has not changed. I find the conclusions to be not adequately supported by the structural data shown here, and these data themselves are quite uneven, and presented and discussed in an unsatisfactory way. Since most of the data have already been published, the necessity of a rapid communication is not evident.

1) In my earlier review, I observed a disconnection between very general claims ("[...] NPD unambiguously show that hydration triggers a transition") and the lack of actual data shown. Such a disconnection still seems to be there. Now two NPD traces (one sample out of all) are shown in fig S7, but there is still no actual NPD refinement, or evidence of a different phase being present in the hydrated vs. dry state beyond all the oxygen positions being full and the protons being in the lattice. The XRD patterns of fig S6, which are fitted to different supercells, look identical down to the pixel level, so the choice of different supercells remains unexplained. Summing up, both the XRD and NPD data shown here do not seem to support the very general conclusions.

2) The authors have now included more details in the Methods section concerning the XAS measurements. However, the Lanthanide M_{4,5} and Co L_{2,3} edges are also mentioned, but they are neither shown nor discussed.

3) The data in fig 2c are adapted from ref 29. However, spectra of BGLC37, 55 and 82 in fig 2e-g also seem to have been published already (doi:10.1039/D2DT03989C) by the same authors: this latter paper however is not cited here, nor is the interpretation concerning spin states, which might be relevant also for the present manuscript. The spectra also look different (BGLC37 peaks at lower energies than both BGLC55 and BGLC82 in this manuscript, but at higher energies in the published paper; BGLC55 is much sharper than the others in this manuscript, but the shapes are all very similar in the published paper). Can the authors comment on the apparently inconsistent presentation and discussion of the spectra?

4) Using a ratio between ionic radii as an x-axis for fig 2b is left unexplained, even in the rebuttal letter. Since there is no apparent relation between hydration and IRR, the choice of IRR is even more confusing.

5) As I already pointed out in my earlier comments, the ionic radii used in fig 3b (and fig S1d) seem to be wrong, and they are not consistent with the source provided by the authors (Jia, J Solid State Chem 1991, 95, 184). Even if Shannon radii are not available for the whole lanthanide series, the dataset by Jia agrees with the Shannon radii to a large extent.

6) As the O K-edge spectra are collected in fluorescence mode, the penetration depth at 530 eV, assuming an atomically flat surface, is on the order of 150 nanometers. A quantitative assessment of how this can be considered as a representative average of the whole sample volume would be good.

7) The authors state that the normalization procedure for XAS follows "the conventional procedure to the edge intensity", but the actual edge intensity is expected to be visible only above ca. 550 eV. Cfr. also doi:10.1039/D2DT03989C, where the same spectra were apparently normalized in a very different way. In any case, showing just a 6 eV range is not an acceptable XAS data presentation. This is a very important point as the authors discuss the absolute intensity of these peaks. The oxygen near-edge spectra must be shown in full in the SI so the normalization procedure can be appreciated, and the peaks due to the interactions with Ln 5d states are also shown.

Reviewer #3

(Remarks to the Author)

It seems that the significance of this work has been improved through the revision stage. The authors clearly responded to all the reviewer's comments, and the revision has significantly enhanced this manuscript. I have no further remarks regarding this manuscript.

Reviewer #4

(Remarks to the Author)

The current manuscript is suitable for publication

Version 2:

Reviewer comments:

Reviewer #2

(Remarks to the Author)

The authors have amended the technical shortcomings in the manuscript concerning diffraction and XAS data presentation.

To the Reviewers,

We herewith submit our revision, with apologies for the long time it has taken, not only due to considerable work in response to the reviewers' suggestions, but also due to a long sick leave by the first author. Only with his return, we have been able to finalise the revision.

Responses to the Reviewers' comments and suggestions

We are grateful for the many insightful reviews and constructive comments, which we have all used to improve the manuscript. Reviewers' comments are reproduced as is and our responses and actions are inserted in boldface.

Reviewer #1:

Overall I have a positive tendency to recommend this paper for publication in Nat. Comm. Nevertheless I also suggest significant revision on various levels.

The topic is highly relevant and suitable for Nat Comm. Mixed conductors which include also proton conductivity are key functional materials for protonic ceramic fuel and electrolyzer cells, which will acquire an increasingly important role in a H₂-based energy system. Several groups find that on one hand, Co-rich materials have a tendency of less pronounced proton uptake, on the other hand Co is typically beneficial for the catalytic activity of such electrode materials. Thus, an improved understanding of a Co-based material as given here is highly welcome. The double perovskites BGLC and related RE-substituted materials have been investigated by the present authors' groups over an extended time. The present paper finalizes these investigations and is able to draw a quite clear picture of a materials system that is really complex. For example, it is found that - even for non-redoxactive RE³⁺ cations - the amount of hydration depends on RE³⁺ being f0, f7 or f14 configuration. And while the amount of proton uptake is quite low, it can still modify the ordering of the cations and propensity for oxidation, which leads to relaxation processes after pH₂O step on multiple time scales.

Therefore the paper is recommended for publication, but in order to be well comprehensible for the readers the presentation of this complicated story needs some improvement.

Presentation:

(1) the paper covers a large range of samples, several data measured also in preceding publications. It would be extremely helpful to collect all lattice parameters, other structural information (space group, presence/absence of A site order etc, quality of the Rietveld fitting) in a table in the SI, indicating also exact state of sample (as-prepared, dry annealing, O₂ or N₂ annealing, hydrated state...)

Response: We agree and have followed the reviewer's advice and expanded Table S1.

(2) the paper shows that TG weight change may originate from hydration, hydrogenation, oxidation, and how the actual proton content can be determined from H₂O/D₂O exchange. Table S1 states "hydration mol% from weight gain in H₂O" - so it seems the numbers there might over/underestimate the real proton concentration? Please add a column giving the real [H⁺] as determined from H₂O/D₂O. Labelling in fig 2b is misleading: axis label says "mol% H" which sounds like real proton conc. (from

H₂O/D₂O) while cation says "weight gain in H₂O" which is sum of all processes. BGLC82 value in fig 2b differs from that in fig 4b

Response: The numbers (mol% H) in fig 2b are taken from the initial (2 h) weight gain of hydration from dry to wet air at 300°C. As can be seen from Fig S4, the initial weight gain is mostly hydration. Since we do not have H₂O/D₂O shifts on all samples, we use the fast initial weight gain to estimate the proton concentration, relying on the experimental verification from D₂O experiments on some of the compositions. The caption of Fig. 2b is edited to explain this better, and we have expanded the elaboration on it in the methods section. The difference for BGLC82 in Fig 2b and 4b reflects the challenges of accurate determination in two different measurements. Figure 4b highlights the difference between measuring isothermal weight changes, reflecting mostly only hydration of an otherwise static structure, and isobaric hydration with decreasing temperature, where the hydration is enhanced by secondary structural changes. This is expanded on in the text. We have added a column in S1 on the D₂O/H₂O results, as suggested.

(3) fig 2b is quite puzzling with the colored arrows. better use just red symbols for "hydrating" and black for "non-hydrating" samples.

Response: We assume the reviewer means Fig. 3b. In that case we agree and have amended the figure according to the reviewer's suggestion.

Fig 3d,e: giving absolute weight changes in mg is not very suitable (magnitude depends on absolute sample weight used). I understand that not always the precise [H⁺] is known owing to different reactions, but at least use "weight change/%" instead of "weight change/mg". Generally, the colored backgrounds in the TG plots (also fig 4) are too strong. Please use lighter colors, and try to simplify the figures as much as possible (i.e. not using color gradients when within such a field the conditions are not changing)

Response: We agree and have amended Figs. 3 d) and e) correspondingly.

(4) fig 4: it would be helpful to insert the sample name abbreviation in each of the plots.

Response: We agree and have marked Figs. 4a and b with BGLC82 and Figs. 4c and d with BGLC37.

Fig 4ab please discuss in more detail in the text that/why the isothermal process (does it proceed by hydration or by hydrogenation, which reaction is assumed for converting weight change into proton conc.) yields lower proton conc. than isobaric (does this imply the isobaric value is overestimated).

Response: Only hydration is assumed for converting weight gain to proton concentration. This will give approximately correct proton concentration for isothermal measurements with short periods in wet atmosphere at each temperature. Hydration is also used as basis for calculation of proton concentrations in the isobaric measurements. In this mode, the material is subjected to wet atmosphere for a longer time, and proton concentrations will be overestimated due to additional oxidation. We see no way to do this differently, and the presentation of the results is meant to highlight the importance of isothermal measurements to get the right result.

Fig 4cd please indicate T. It would be nice to directly indicate in this fig.

Response: We agree and have updated the figure with information of temperature (300 °C).

which fraction of H is from hydration and which from hydrogenation.

Response: In Figs 4c and d we don't give fractions of H. In oxidizing conditions, the weight gain (mg) is a combination of hydration and oxidation. The additional weight gains in D₂O give the total mass of H, which are converted to mol% from the total sample mass. Under inert conditions we do the same, and the total mass of D again allows us to calculate mol% H. The ratio between mass gain in D₂O vs H₂O is much larger in N₂ atmosphere, revealing that most of the protons come from hydrogenation.

Fig. 4e please indicate T, that it is in N₂, and what are the p_{H2O} and p_{D2O}.

Response: We have improved the caption for 4e. p_{H2O} for each step is given in Fig.4f. T is 300°C.

(5) fig 5 middle part: please write "dry" and "hydrated" close to the respective structures. Bottom part: the small coordinate system with axes labelled p_{O2} and p_{H2O} is easily overlooked. Please indicate more prominently that horizontal axis refers to proton concentration, and vertical axis to oxygen uptake (oxidation). Please indicate more clearly that the "hydroxidation" happens only because the hydration leads to A site disorder which then favors further oxidation (without this it is not conceivable why hydration and oxidation should occur in parallel since they both consume oxygen vacancies)

Response: We have improved the figures as suggested and clarified in the associated text.

Additional discussion/explanation:

(6) within the presented data set there is a strong correlation that only f₀, f₇ or f₁₄ RE³⁺ materials show large hydration. However, based on the usual "textbook knowledge" that f orbitals have a very small radius and are thus "quite inside the atom" it is not so obvious how the details of f occupancy (6 or 7 or 8 electrons) would affect the t_{2g} orbitals of Co which at approx. 3.5 Angstrom distance. Might it help to think about Jahn-Teller-like effects at RE which could modify RE-O bonding, or generally how f occupancy could affect RE-O bonds (and this then the Co-O bonds)? Along this line, maybe look also at RE-O bond lengths and angles (similar to Co-O in fig A1ab)?

Response: The reviewer's comment is pointing to an important alternative causing the f-shell influence on the Co-O bond. We have revisited our structural data to look at the RE-O bond and see if there is any link to f₀, f₇ and f₁₄. We did not find any clear trend, and figures of bond-lengths for RE-O and bond-angles for Co-O as a function of Ln radius are now added as Figure S1 c and d, respectively. This does not, of course, rule out a Jahn-Teller-like effect from the RE-O bond, but our study does not support any conclusion on this. We have, however, included reference to a recent publication where a Ln4f-TM3d hybridization is reported for a perovskite structure. We have made a comment on this in the text.

(7) fig 2d-h: it is impressive but to some degree also astonishing how a small proton uptake of about 1% (0.01 H per AA'B₂O₆ !) strongly modifies the intensity of the O-prepeak (which is averaging over all O's, most being unprotonated). Any further explanation is welcome

Response: It is indeed intriguing. In our opinion this may be related to the degenerate band structure and how small changes affects the configuration of electrons. It seems that the O-K pre-edge structure is highly sensitive to even small changes in protonic defect concentration. The ability to incorporate protonic defects, being govern by the electronic structure of O 2p-Co 3d hybridized orbital, manifests itself with changes in the averaged signal. Since we do not have a reliable answer to the question, we have decided to avoid speculations in the manuscript, and left this issue un-commented.

(8) Co is known to have several spin states. What is known about spin of Co in present samples, it may also affect nature of Co-O bond and thus tendency for protonation? maybe add some information in the SI

Response: Our previous studies shown that the mixture of Co³⁺ HS, and Co⁴⁺ LS is present in the investigated materials, however it is difficult to calculate the concentration of cobalt ions at each oxidation and spin state on the basis of only XAS results. The correlation between cobalt spin state and tendency for protonation was not observed, since no significant changes in Co spin state were revealed for hydrating and non-hydrating materials. However, fully LS cobalt is not present for any of the studied materials contrary to e.g. in the LaCoO₃ system [ref. Dalton Transactions 52 (17), 5771-5779; Materials 13 (18), 4044]

(9) hydration vs hydrogenation are quantified from the H₂O/D₂O experiment. However, even when pH₂O=pD₂O are used, the hydration mass action constants for H and D might slightly differ because zeropoint vibration effects. At least mention that this might add a little uncertainty

Response: The effects suggested are expectedly negligible here in view of the qualitative approaches in this article, but we have improved the description of actual levels of H₂O and D₂O in the gases in the Methods section and throughout the text and – to meet the reviewer’s suggestion – added a sentence on the uncertainties stemming from multiple minor effects in isotope exchange experiments.

Further points:

- fig 1b please write "ion radius ratio" instead of "IRR"

Response: Implemented.

- caption fig 2 there is no "g-k", should be d-h?

Response: Thanks for spotting the mistake. Corrected.

- fig 3b: authors seem to use a different scale for ionic radii than in [20] - why? the radii in [20] are the wellknown Shannon radii, with 1.16 Å for La in CN=8

Response: We use <https://www.sciencedirect.com/science/article/abs/pii/002245969190388X> as ref on ionic radii. This gives us a full set of 12-coordinated Ln radii. We use this in a follow-up manuscript (in preparation) on stability in steam. Shannon radii don't give values for the whole matrix in 12-coordination.

- ion implantation is not a standard lab technique. Thus it must be specified how/here the D implantation as SIMS standard was done

Response: We have described the ion implantation for fabrication of the SIMS reference standard by including the following paragraph:

“A SIMS reference standard was prepared by ion implantation using a 1MeV Tandem ion accelerator from National Electrostatics Corporation (NEC). Deuterium ions were extracted from a gas source and accelerated to an energy of 200 keV and implanted into a separate sample to a fluence of 5×10^{14} cm⁻². The sample was used as a reference sample to convert the measured deuterium signal into absolute concentration. The SIMS reference sample was measured with the same instrument parameters as the sample of interest, utilizing the implantation fluence to calculate the conversion factor, i.e. the relative sensitivity factor of deuterium. “

- pg 9 line 225 should probably read "hydration causes A site disordering"?

Response: Thanks for pointing out this serious mistake and, hence, the general inconsistency of that paragraph. The text in that paragraph has been corrected accordingly.

- lines 263 and 267 please mention the figure numbers which show the respective correlations
Response: Done.

- fig S6 maybe indicate more clearly which are the specific reflections that evidence the A site ordering. maybe use log scale for intensity to give more weight to small reflections

Response: There are no reflections that evidence A-site ordering. The two Bragg-patterns are the same and can be fitted to both dry and hydrated state. Our conclusions on A-site disorder comes from having studied more than 30 spots and grains with STEM-EDS without finding A-site order in the hydrated material. Refinements to a 2a3b3c supercell is the confirmation that unifies XRD and STEM-EDS findings.

- standard sample preparation is 1150°C for 48h - what is resulting relative density. Does this directly yield sufficiently dense samples for the SIMS experiment, or were they made by modified process?

Response: The pellets for SIMS were sintered at 1190°C for 10 hours to obtain sufficient relative densities for SIMS measurements. We have added this information.

- I guess "powder" for TG mentioned in line 340 is obtained by crushing the 1150°C for 48h pellets ? then please write that. Any idea of resulting grain/particle size?

Response: The powders were indeed from crushed and hand milled pellets, the average grain sizes haven't been measured, but the BET measurements showed that the surface area was in the range of 1-5 m²/g reflecting that the powders were coarse enough to represent bulk properties.

Reviewer #2:

This paper studies Co-based double perovskites with mixed A2+ and A3+ cations in the A-site. It employs a variety of structural, thermogravimetric and electrochemical measurements to draw general conclusions about the hydration behavior.

Although the manuscript contains some interesting techniques, I found it to be unorganized in the presentation of data and results, and this regrettably influenced clarity. In general, the discussion and conclusions are insufficiently supported by the results. In addition, the discussion frequently resorts to data that was already published, or even not shown.

I recommend that this paper is submitted to a more specialized journal after its several problematic aspects are amended. Specific points follow.

1) Most figures are assembled out of different panels, which can help condensing information, but then the meaning of the large Figure 1 is unclear.

Response: We are not sure how to respond to this comment. We have left Figure 1 as it is, given its importance in giving a detailed overview of occupancies for cations and oxygen, and also for labelling of Co sites. We believe the reader will benefit from having a large structure to consult when reading the manuscript.

2) The authors mention that both neutron (NPD) and synchrotron (SR-XRD) powder diffraction have been used to "unambiguously show that hydration triggers a transition into an oxidised A-site disordered" supercell (page 11). I could not find NPD in either the manuscript or the Supporting information. On the other hand, page 1 only mentions SR-XRD as the technique to refine the position of seven different oxygen sites.

Response: We have used NPD refinements as starting point for refining SR-PXD results on some selected materials. Since our hydrating materials generally contain various amounts of Gd, which is a neutron absorber, we had to make separate ¹⁶⁰Gd enriched samples especially for NPD. Hence, the quanta are small and different samples were used for NPD and SR-PXD, where we focused on the hydrated and non-hydrated samples without ¹⁶⁰Gd enrichment. However, in the particular case of BGLC37, we performed NPD and SR-PXD on both dry and D₂O-exposed ¹⁶⁰Gd enriched sample. We have inserted a figure in SI with examples of NPD results for BGLC28 and for BGLC37 in dry and wet (D₂O) atmospheres.

3) The abstract anticipates that hydration is linked to A-site basicity, but this is actually never discussed directly.

Response: We discuss it in the introduction of the manuscript. Based on the reviewer's comment, we now bring it in more explicitly also in the Discussion.

4) One of the main hypotheses put forward is that closed or semi-closed 4f states (La³⁺, Lu³⁺, Gd³⁺) influence hydration, and the authors use open/closed symbols in Fig 2b and Table S1. There seems to be inconsistency here: why is BGLYC721 marked as closed shell but BGLuC721 is not? Why are BGLSrC(C) or BGLuC721 not closed shell? The trend seems less clear than it is presented by the authors, as very similar compositions show markedly different hydrations.

Response: BGLSrC has Sr substituted for Ln, and it is thus not relevant to label it closed-shell. It is also cubic. BGLLuC622 and BGLLuC721 are closed-shell. We have corrected this accordingly in the figure and table.

5) Fig 2a is mostly taken from ref. 20. Fig2b is plotted against IRR as the x-axis: what is the meaning of the radius ratio and why should hydration be plotted against it? This is never explained. In Figure 3b the x-axis seems to be way too low, and I suspect the ionic radii for an octahedrally coordination were incorrectly considered.

Response: Yes, 70 % of the data in Fig 2a is from [20]. We have inserted a comment and reference on this in the text. Regarding Fig 2b: There is no other relevant x-axis, at least we don't have another suggestion for x-axis. The ratio between the averaged ionic radii for the cations on A^I and A^{II} sites is of interest for structural and stability reasons. Since there is no other relevant x-axis, we hope for accept to keep it as it is. In fig.3b we use tabulated values for 12-coordinated Ln from <https://www.sciencedirect.com/science/article/abs/pii/S002245969190388X> as ref on ionic radii (see comment also for Reviewer #1.)

6) As the authors underline how the extensive composition matrix (page 2) supports the correlations, one would expect that the XAS spectra on Fig 2c-h would cover the same samples as in Fig 2b, while it only Gd/La compositions are shown.

Response: We only chose examples of hydrating (BLC, BGC, BGLC37 and BGLC82) and non-hydrating (BGLC55, BPC and BNC) due to limited beam-time.

Also since the absolute intensity of XAS peaks is discussed, some more information about measurements (is it electron yield? Are surface states expected to be the same as the bulk?) and data reduction (how are the spectra normalized?) would be good. Also panels c-e and f-h are shown with a common x-axis, but the tickmarks don't align.

Response: The presented data were collected in the XAS fluorescence mode, thus it refers not only to the most outer layers of the materials, but also to the volume of samples. The data were normalized according to the conventional procedure to the edge intensity after subtracting the background signal fitted with the first-order polynomial fitted to the pre-edge regime. This information was added to the experimental section. The tick-marks shift in the figure 2 resulted from the slight changes in the graphs scale. The error was corrected. What is more, the line types used in figure 2c were changed to be more distinguishable in a monocolour printed version.

7) At least laboratory XRD with Rietveld refinements should be shown for each sample in the Supporting information. The only diffraction data actually shown is Fig S6, which is too small to appreciate any difference between the two panels.

Response:

We have expanded table S1 with structural information for all samples.

There is also no goodness of fit or statistical significance reported in order to tell the two models apart. Previous data by the same authors (refs. 20 and 28) is much better presented in comparison.

Response: We have inserted statistical data for all refinements in table S1.

Reviewer #3:

This study presents the governing principles of protonation by systematically analyzing the hydration trends of cobalt-based double perovskite, $A^I A^{II} Co_2 O_{(6-\delta)}$. The authors report that lanthanide A-site dopants with a closed or semi-closed Ln 4f shell (i.e., Ln = La, Gd, Lu) induce electron hole transfer from eg to t_{2g} in the Co-O bond, as supported by XAS measurements. Additionally, STEM-EDS, SR-PXD, and PND clearly demonstrate that hydration triggers a transition to an oxidized A-site disordered 2a3b3c Pmmm supercell, which affects both oxidation and hydration equilibria. This work is well-designed, and the analysis is meaningful, particularly in designing mixed proton-electron positrodes that maximize hydration and surface reaction kinetics. Such advances could offer substantial contribution to the PCEC community. Therefore, I would recommend that this paper is suitable to be published in this journal after revision with following comments.

- In Figure S3, the concentration profile of BGLC37red with D₂O exchanged at 600°C shows a distinct pattern of initially increasing, then decreasing, and finally reaching saturation. This behavior is not mirrored in BGLC82. Could the authors elaborate on the underlying reasons for this unique trend in BGLC37red?

Response: Previous TEM studies have revealed a certain segregation of Ba cobaltite or Ln oxides or hydroxides to the surfaces or sub-surfaces. These segregations depend on composition (Gd/Ln), temperature and atmosphere. Hence, any differences in the proton concentration profile from surface to bulk between compositions may arise from cation concentration gradients. We have decided to leave this issue for further studies. We value the comment and agree on its importance. Nevertheless, we prefer to leave this topic out from the text.

- The authors state that the hydrated supercell is oxidised and contains oxygen sites like O10, which are more easily protonated than O4 and O6 in the a2b2c phase, leading to enhanced hydration and oxidation in the 2a3b3c structure. Could the authors provide further explanation or references to support this assertion? Understanding the specific reasons why O10 sites are more susceptible to hydration compared to O4 and O6 would clarify this aspect of the study.

Response: This is a good point. We base this statement on previous studies that show a more complete TM-O electron transfer when the bond angle deviates from 180° and the bond is elongated. We have made an entry on this and inserted references.

Oxidation is a result of – or causing – the phase transition itself. In the a2b2c phase, oxygen vacancies compensate for the A^I vs A^{II} size mismatch. In the 2a3b3c phase there is no difference between the A-sites, and hence no need for oxygen vacancies.

- Regarding a STEM-EDS results as provided in Figure 5a, the authors mentioned that the ordering structure in BGLC37 become disorder after hydration. I'm simply wondering if this phenomenon may be reversed. Does hydrated BGLC37 back into an ordered structure after dehydration? Furthermore, do you have a change to check the chemical stability under hydration conditions?

Response: We have tried to reverse the ordering, but it has not been successful. Dehydration / reduction is much slower than hydration / oxidation, so it may be a matter of sufficient time. We have a manuscript under preparation on chemical stability for the whole composition matrix in steam.

- In figure 2a, I think there are some missing TGA results. Please show all TGA data with all compositions even in supplementary figure.

Response: We don't show all TG curves as it would be too chaotic. In 2a we only show examples. All results are in Table S1 and Fig. 2b.

- Why the mass change of BLC with cubic structure and BLC with ordered structure is more than 4 times different. Is it because the unit cell volume has increased? Please clearly explain this point.

Response: The four-fold increase in mass gain for ordered BLC reflects the different electronic structure and different oxidation state for the two polymorphs. The more reduced and ordered structure hydrates more because the more reduced Co ions transfer fewer holes to O. This phenomenon is indirectly supported by the findings in Fig d-h, where we show how depopulation of electron holes in the Co_{t2g} orbitals correlates with susception to protons in the Co-O surroundings. The example we give in Fig. 2a is supported by the same effect for reduced and oxidised BGLC37 as shown in Fig. 3d and e. We have clarified with an entry in the manuscript.

- Based on the TGA data in Figure 2a, fully Gd and La doped in BaLnCo2O6-d samples seem to exhibit the highest mass change under both dry and wet conditions. Nonetheless, Gd and La co-doped sample, BGLC, is used as representative material in this work. There is a lack of clear explanation regarding this.

Response: Even though BLC and BGC gain more mass in wet atmospheres, we believe this is mostly due to oxidation and that proton concentrations are small. H2O/D2O shifts may confirm this, but we don't have them. One can also see that BLC and BGC maintain all the mass gained in wet conditions after reverting to dry. Finally, XAS measurements of BLC and BGC in dry and wet sates indicate less repulsion on electron holes from protons in these compositions, indicating lower proton concentrations than in the BGLC compositions. Hence, we interpret the weight gain in wet atmosphere as mostly oxidation. Furthermore, both fully La- and Gd-doped compositions are unstable in wet atmospheres: Orthorhombic BLC reverts to cubic (oxidised) structure in wet oxidising conditions, while BGC decomposes under water pressures relevant for its potential use in proton ceramic cells. The introduction of mixed La/Gd composition stabilises the ordered structure while maintaining the compositional integrity in high steam pressures. This is, however, the scope of our next publication which is currently under preparation. We have made an entry in the manuscript regarding BLC(O) and BGC.

Minor points:

- There are errors in Figure 2. In caption b), "See also Table S2 " need to be changed "See also Table S1". In figure 2b, there is duplicate data point with an IRR close to 1.15 (BSrPC). Please check it.

Response: Thanks for spotting these – they are corrected.

- There are typos on page 6. In figure 3b, y1-axis is Co oxidation state and y2-axis is oxygen non-stoichiometry. And blue arrows indicate hydrating and red arrows indicate non-hydrating in figure 3b. It seems to be incorrectly labeled, unlike the manuscript. Please revise these errors.

Response: Thanks for spotting this collection of our errors. We have corrected them in the text.

Reviewer #4:

This work presents a new governing principle for the hydration and proton transport in Co-based double perovskites. This principle is extracted from a comprehensive experimental campaign that includes the preparation of a large material library (experimental matrix based on Ln combinations) and fundamental characterization looking at the atomic-level structural and electronic aspects together with proton-transport methods.

The quality and completeness of the presented results are of the highest quality. The work is generally well described, and the discussion is well articulated and succinct. I am convinced that this article will impact the research community working on proton ceramics and electrochemistry. I recommend the publication of the paper subjected to minor revision, taking into account the following list of comments and suggestions:

1- Fig. 2b. This chart is difficult to read, and I needed to go to Table S2 (back and forward) a couple of times. Is there a simple way to include the composition information directly in Fig. 2.

Response: We don't know how to make it clearer. We have tried to limit the numbering and labelling in the figure and left it as an overview rather than a source of data. However, we have expanded table S1 with more information on all samples investigated.

2- Page 6. When comparing proton concentrations in BGLC82 and BGLC37, it is properly concluded that there is a dominance of Ln basicity over Co oxidation state. For the general reader, I recommend the authors explain more in detail the term 'Ln basicity' in this context and its implications in the proton and electronic transport.

Response: We have emphasised this aspect – with Ln basicity based on electronegativity – in the manuscript. The reviewer mentions transport, but the relevant discussion here deals with thermodynamics – not transport – of defects.

3- Last paragraph, page 6. Please, use a unique symbol for deuterium. Now, D and $2H$ are used simultaneously in the text, Fig. 3 and Fig. S3

Response: Thanks. We have made sure it is consistent. 2H is used for the hydrogen atomic isotope in the SIMS results, while D_2O is used for deuterium oxide as is the standard abbreviation. These two abbreviations are the ones most commonly used for the two purposes, and keeping them makes the manuscript easier to read.

4- Page 9. There is a very interesting conclusion: "the two lanthanides are sufficiently small to maintain the same oxidation equilibrium". Please, could you elaborate more on this and extrapolate to other combinations and systems, i.e., as a general principle?

Response: It means that the TG profile is fully reversible for BGLuC. Hence, no oxidation or structural changes and no slow mass gain in wet conditions. We have elaborated on it a bit more but have not been able to extrapolate to other systems or a general principle.

5- Fig. 5. I suggest the authors label each individual image and re-arrange it to improve the overall readability of the figure. In addition, as a general conclusion of the figure, it is stated, "We note that pure hydration in mixed valence oxides like the ones we consider here occurs only under specific combinations of pO_2 and pH_2O ." Can the authors be a bit more specific on the type of combinations

pO₂-pH₂O enabled hydration and the associated materials properties that determine this 'hydration space'.

Response: We have labelled each part. The conclusion is based on the measured hydroxidation and hydrogenation in oxygen and N₂, respectively, for BGLC37 red(uced). The combination of pO₂ and pH₂O is dependent on the average Ln radius and how resistant the structure is to A-site disorder. We have added a sentence on this in the manuscript, and the issue is also treated in the elaborated comment on BGLuC (reviewer comment 4 above) in the manuscript.

6- To illustrate the potential and versatility of the proposed principles, the authors could comment on the potential application of these principles to the design and understanding of other mixed oxides exhibiting hydration and/or proton transport, if possible, providing specific examples of composition and crystalline structures. Can this be connected with disorder or entropic effects in different classes of oxides?

Response: The most important consequence of these findings is how the TM-O bond can be engineered to adjust the population of electrons and holes. This means adjusting the TM-O charge transfer, and thus the ionicity of the bond. The secondary – or less detrimental – governing principles state that Ln basicity is more important than Co oxidation state, as illustrated by the higher proton concentration in the more oxidised, but also more basic BGLC37 than in the more reduced, but less basic than BGLC82 and the even more reduced and even less basic BGLuC82. We also confirm the importance of bond-angles, as has previously been only theoretically determined. It is difficult for us to suggest possible structures or compositions here, but we have added a sentence in the text on the practical importance of the findings.

Responses to reviewers #1, #3, and #4:

These reviewers recommended the manuscript revision 1 to be published without comments.

Response to reviewer #2

Reviewer #2 general comment: In the response letter and revised paper, the authors have partially addressed my earlier comments. However, my general assessment of this paper has not changed. I find the conclusions to be not adequately supported by the structural data shown here, and these data themselves are quite uneven, and presented and discussed in an unsatisfactory way. Since most of the data have already been published, the necessity of a rapid communication is not evident.

Reviewer #2 comment 1): In my earlier review, I observed a disconnection between very general claims ("[...] NPD unambiguously show that hydration triggers a transition") and the lack of actual data shown. Such a disconnection still seems to be there. Now two NPD traces (one sample out of all) are shown in fig S7, but there is still no actual NPD refinement, or evidence of a different phase being present in the hydrated vs. dry state beyond all the oxygen positions being full and the protons being in the lattice. The XRD patterns of fig S6, which are fitted to different supercells, look identical down to the pixel level, so the choice of different supercells remains unexplained. Summing up, both the XRD and NPD data shown here do not seem to support the very general conclusions.

Reply: We agree that structural data alone does not give an unambiguous conclusion on the structural transition. It is the combination between STEM-EDS elemental mapping and XRD / NPD that allow us to make such a bold statement. It has been challenging to arrive to this conclusion given a clear Ba-Ln order-disorder transition on the A-site as shown by STEM-EDS and at the same time no change in the XRD diffractogram upon hydration. Seeing that a disordered supercell gives the same Bragg reflections offer a logical and sound explanation to both of these observations.

Regarding the number samples characterised by NPD, we here only present two – BGLC37 and BGLC28. The reason for this is the sample cost, as these samples must be made with a non-neutron absorbing ¹⁶⁰Gd isotope. None of these results are previously published, and since we believe the hydroxidation reaction to be a general reaction for mixed conducting electrode materials, we find the results highly relevant and worthy of rapid communication.

Having said that, we fully understand the reviewer's comment regarding NPD – not only SR PXD – refinements to support our claim. We have therefore also included refinements of NPD on the dry and hydrated BGLC37 isotope sample. The refinements show a slightly – maybe insignificantly – better fit to the disordered 2a3b3c supercell with Rwp value slightly lowered for both dry and hydrated samples as compared to refinement to a2b2c orthorhombic structure. Figure S7 is now modified to include NPD refinements results. We have also included the NPD refinement results in S.I.

Reviewer #2 comment 2): The authors have now included more details in the Methods section concerning the XAS measurements. However, the Lanthanide M_{4,5} and Co L_{2,3} edges are also mentioned, but they are neither shown nor discussed.

Reply: We apologize for the inconsistency in the experimental description. The information on the other edges has been removed from the manuscript.

Reviewer #2 comment 3): The data in fig 2c are adapted from ref 29. However, spectra of BGLC37, 55 and 82 in fig 2e-g also seem to have been published already (doi:10.1039/D2DT03989C) by the same authors: this latter paper however is not cited here, nor is the interpretation concerning spin states,

which might be relevant also for the present manuscript. The spectra also look different (BLGC37 peaks at lower energies than both BGLC55 and BGCL82 in this manuscript, but at higher energies in the published paper; BGLC55 is much sharper than the others in this manuscript, but the shapes are all very similar in the published paper). Can the authors comment on the apparently inconsistent presentation and discussion of the spectra?

Reply: The data presented in doi:10.1039/D2DT03989C and this article were measured in different projects under different grants and under non-identical conditions. Data presented in the above-mentioned paper were measured in TEY mode and under a narrower energy range (510-570 eV) (Fig. 1 below), which covers mostly the pre-edge features of the spectra. Consequently, these data could not be normalized to the edge regime, but we decided to conduct the normalization to the pre-edge feature around 528 eV. We are aware that with this approach we lost the information on the relative intensities of the spectra, but we decided to conduct a quantitative analysis of pre-peak position, which is related to the Co oxidation and spin state. What is more, since the data published in doi:10.1039/D2DT03989C were acquired in the TEY mode they differ from the data presented in this work.

Figure 1 XAS spectra in the whole measured energy range, presented in doi:10.1039/D2DT03989C.

For this study, we used the FY mode, which is more representative for the bulk properties than the TEY mode. We also expanded the energy range to 475-750 eV, which enables us to normalize data to the edge regime. The acquisition mode was also changed to FY. See Fig. 2 below. This approach gives information not only on the present spin and oxidation state but also the relative density of unoccupied states. Furthermore, this study aimed to show the difference between the material before and after exposure to a humidified atmosphere. For this purpose, the samples were initially dried at 900 °C for 24 h and then exposed to water vapour. The spectra of dried and humidified materials are presented in this manuscript. In the previous studies, as-synthesized materials were characterized. We elaborated on materials preparation in the experimental section. The difference in both sample preparation and measurement conditions is the reason for the observed differences between the two data sets.

Figure 2 XAS spectra in a wider energy range

Reviewer #2 comment 4): Using a ratio between ionic radii as an x-axis for fig 2b is left unexplained, even in the rebuttal letter. Since there is no apparent relation between hydration and IRR, the choice of IRR is even more confusing.

Reply: We believe we answered this comment in the previous rebuttal letter. We there replied “There is no other relevant x-axis, at least we don't have another suggestion for x-axis. The ratio between the averaged ionic radii for the cations on A^I and A^{II} sites is of interest for structural and stability reasons. Since there is no other relevant x-axis, we hope for accept to keep it as it is.” We have at present no further explanation to offer and again think it should be left as it is.

Reviewer #2 comment 5): As I already pointed out in my earlier comments, the ionic radii used in fig 3b (and fig S1d) seem to be wrong, and they are not consistent with the source provided by the authors (Jia, J Solid State Chem 1991, 95, 184). Even if Shannon radii are not available for the whole lanthanide series, the dataset by Jia agrees with the Shannon radii to a large extent.

Reply: We are embarrassed, the reviewer is right. The x-axis values for average *L_n* radius were indeed wrong and not according to our reference. The error is now corrected and figure 3b shows the correct values for average *L_n* radii.

Reviewer #2 comment 6): As the O K-edge spectra are collected in fluorescence mode, the penetration depth at 530 eV, assuming an atomically flat surface, is on the order of 150 nanometres. A quantitative assessment of how this can be considered as a representative average of the whole sample volume would be good.

Reply: The two basic modes of acquisition in X-ray absorption are TEY and FY. The latter, used in this work, is the most suitable for assessing the volume properties of the material. A 150 nm penetration depth is far deeper than surface space charge layers (and accompanying segregation) in such a material with high defect and conductivity levels. It is also far deeper than expected diffusion would reach at ambient temperatures during storage and transfer of samples prior to XAS. We equilibrated the samples in either dry or wet conditions before the measurement and may expected on basis of the above that deviations near the surface play a minor role in XAS/FY compared to the bulk.

Reviewer #2 comment 7): The authors state that the normalization procedure for XAS follows "the conventional procedure to the edge intensity", but the actual edge intensity is expected to be visible only above ca. 550 eV. Cfr. also doi:10.1039/D2DT03989C, where the same spectra were apparently normalized in a very different way. In any case, showing just a 6 eV range is not an acceptable XAS data presentation. This is a very important point as the authors discuss the absolute intensity of these peaks. The oxygen near-edge spectra must be shown in full in the SI so the normalization procedure can be appreciated, and the peaks due to the interactions with Ln 5d states are also shown.

Reply: As described in detail in the answer to comment no 3, we agree with the Reviewer that the actual edge intensity is visible at energies higher than 550 eV. The data presented in this manuscript were normalized to the post-edge intensity level (550-570 eV), after background subtraction. We inserted the spectra in the wider energy range in the SI.